# Outer membrane vesicles from a mosquito commensal mediate targeted killing of *Plasmodium* parasites via the phosphatidylcholine scavenging pathway

Han Gao [1,2,3], Yongmao Jiang [1,2,3], Lihua Wang [1,2,3], Guandong Wang[1,2], Wenqian Hu [1,2], Ling Dong[1,2] & Sibao Wang [1,2] ✉

The gut microbiota is a crucial modulator of *Plasmodium* infection in mosquitoes, including the production of anti-*Plasmodium* effector proteins. But how the commensal-derived effectors are translocated into *Plasmodium* parasites remains obscure. Here we show that a natural *Plasmodium* blocking symbiotic bacterium *Serratia ureilytica* Su_YN1 delivers the effector lipase AmLip to *Plasmodium* parasites via outer membrane vesicles (OMVs). After a blood meal, host serum strongly induces Su_YN1 to release OMVs and the antimalarial effector protein AmLip into the mosquito gut. AmLip is first secreted into the extracellular space via the T1SS and then preferentially loaded on the OMVs that selectively target the malaria parasite, leading to targeted killing of the parasites. Notably, these serum-induced OMVs incorporate certain serum-derived lipids, such as phosphatidylcholine, which is critical for OMV uptake by *Plasmodium* via the phosphatidylcholine scavenging pathway. These findings reveal that this gut symbiotic bacterium evolved to deliver secreted effector molecules in the form of extracellular vesicles to selectively attack parasites and render mosquitoes refractory to *Plasmodium* infection. The discovery of the role of gut commensal-derived OMVs as carriers in cross-kingdom communication between mosquito microbiota and *Plasmodium* parasites offers a potential innovative strategy for blocking malaria transmission.

Vector control remains the mainstay of current malaria control programs[1]. However, the increasing incidence of mosquito insecticide resistance[2], changes in mosquito behavior to more outdoor and/or early biting[3], as well as malaria parasite resistance to drugs[4] have stalled the progress against malaria in recent years[5]. Therefore, new strategies to control the disease are urgently needed. Malaria parasite transmission depends on the successful completion of a complex developmental program in the lumen of the mosquito midgut, where malaria parasites suffer severe losses[6], making the mosquito midgut a prime target for intervention. A promising approach for malaria control is not to kill mosquitoes, but instead to interfere with parasite transmission by targeting the mosquito midgut stages of parasite development[7].

The gut of adult mosquitoes is colonized with a complex community of commensal microbes, known as the gut microbiota, which

[1]CAS Key Laboratory of Insect Developmental and Evolutionary Biology, CAS Center for Excellence in Molecular Plant Sciences, Shanghai Institute of Plant Physiology and Ecology, Chinese Academy of Sciences, Shanghai, China. [2]CAS Center for Excellence in Biotic Interactions, University of Chinese Academy of Sciences, Beijing, China. [3]These authors contributed equally: Han Gao, Yongmao Jiang, Lihua Wang. ✉e-mail: sbwang@cemps.ac.cn

plays a crucial role in host physiology, particularly in modulation of pathogen infection[8]. Malaria parasite and the gut microbiota share the same compartment – midgut – where the most vulnerable stages of the *Plasmodium* development occur in mosquitoes[7]. An increasingly attractive strategy for arresting parasite development is to populate mosquitoes with anti-*Plasmodium* symbiotic bacteria[9]. This strategy, known as paratransgenesis or symbiont-based transmission blocking strategy, has shown promise[7]. Engineered gut bacteria expressing anti-*Plasmodium* effectors[10] and more recently, a naturally occurring symbiotic *Serratia ureilytica* Su_YN1 bacterium[11], were shown to be able to spread through mosquito populations while driving mosquito refractoriness to *Plasmodium* infection. *S. ureilytica* Su_YN1 inhibits *Plasmodium* development via secretion of a potent anti-*Plasmodium* lipase protein AmLip, which is transported into the *Plasmodium* parasites, disrupting them via its lipid-hydrolyzing activity[11]. However, the mechanism by which anti-*Plasmodium* effector proteins including AmLip are translocated into *Plasmodium* parasites remains totally unknown.

Mosquito gut symbiotic bacteria are mainly gram-negative bacteria, which widely utilize type I secretion systems (T1SS) to deliver effector proteins to extracellular spaces[12]. In paratransgenesis, T1SS was employed to secret various anti-*Plasmodium* effector proteins to inhibit *Plasmodium* parasite cells in the mosquito midgut[7,10,13]. However, T1SS secreted effector proteins are not directly transported to target cells and not protected after released. Gram-negative bacteria have also evolved specialized protein secretion systems, such as T3SS, T4SS and T6SS, which translocate bacterial effector proteins into host cells following direct contact[14]. In addition, many pathogenic bacteria release outer membrane vesicles (OMVs) that act as a T0SS delivery system to transport bioactive molecules to target cells[15]. However, to our knowledge, no examples of gut commensal bacteria-derived extracellular vesicles have been described for delivering molecules to gut pathogens.

Here we show that host blood-meal serum strongly induces the *Plasmodium*-inhibiting gut commensal bacterium *Serratia* Su_YN1 to release OMVs and the antimalarial effector protein AmLip. The secreted AmLip is preferentially loaded onto the OMV surface and transported to *Plasmodium* parasites. Notably, Su_YN1 OMVs incorporate certain serum-derived phospholipids that are important for OMV uptake by *Plasmodium* parasites via the phosphatidylcholine scavenging pathway, uncovering a novel mechanism by which gut commensal bacteria deliver effector molecules via OMVs to selectively attack the parasites and thus render mosquitoes refractory to malaria parasite infection.

## Results

### The commensal-derived antimalarial effector protein AmLip preferentially accumulates on OMVs

We recently reported that the symbiotic bacterium *S. ureilytica* Su_YN1 directly inhibits *Plasmodium* development via secretion of antimalarial lipase AmLip[11]. To investigate how AmLip secreted from Su_YN1 reaches the parasite in the mosquito gut, we sectioned the midgut of blood-fed mosquitoes carrying Su_YN1 and examined the distribution and localization of AmLip by immune electron microscopy (IEM), using specific AmLip mouse antiserum we previously prepared (Supplementary Fig. 1a, b)[11]. Unexpectedly, AmLip-stained gold particles appeared to predominantly accumulate on membrane vesicle-like structures in the mosquito midgut (Fig. 1a). The lack of the exosomal marker CD63 and CD9 positive staining and the presence of AmLip positive staining indicate that these vesicle-like structures are not of eukaryotic origin (Supplementary Fig. 1c, d). Recent studies indicated that bacteria may release specialized outer membrane vesicles (OMVs) in response to certain conditions[16]. We examined membrane vesicles produced by Su_YN1 in the gut by transmission electron microscopy (TEM) (Fig. 1b, upper panel) and scanning electron microscopy (SEM)

(Fig. 1b, lower panel). Remarkably, massive OMVs were released by *S. ureilytica* Su_YN1, which are budding from the bacterial outer membrane and scattered nearby.

Digestion of blood meal imposes stress on the mosquito gut bacteria[17]. To test whether Su_YN1 OMV production was specifically induced by host blood in mosquito gut, *Anopheles* mosquitoes carrying Su_YN1 were fed with either a sugar or a blood meal and sectioned for TEM observation. Interestingly, Su_YN1 OMV production could only be detected in blood-fed but not sugar-fed mosquito guts (Fig. 1c), indicating that host blood components induce the production of Su_YN1 OMVs.

### Host serum induces the commensal *Serratia* Su_YN1 to secrete OMVs

OMV production may be induced in response to various environmental conditions[16]. To examine which blood component induces Su_YN1 OMV production, we fractioned blood components as total blood, serum, and RBC lysate. These components were co-cultured with Su_YN1 and culture supernatants were examined by TEM. We found that host serum strongly induces OMV production while RBC lysate does not (Supplementary Fig. 2a). Moreover, various animal and human sera strongly induced Su_YN1 OMV production (Supplementary Fig. 2b, c, d). We used fetal bovine serum (FBS) in subsequent experiments due to its commercial availability and consistent quality. To confirm whether serum indeed induces Su_YN1 OMV production in the mosquito gut, we fed Su_YN1-carrying mosquito with EV-free FBS and then detected OMVs in the mosquito gut using TEM. OMVs were only observed in the guts of Su_YN1 carrying mosquito fed with FBS, but not axenic mosquitoes fed with FBS, or in Su_YN1 carrying mosquitoes fed with a sugar meal (Supplementary Fig. 2e).

We next cultured Su_YN1 with or without FBS and quantified OMV yield. The FBS was pre-cleared by ultracentrifugation to deplete native extracellular vesicles (Supplementary Fig. 2f). In response to serum induction, Su_YN1 bacterium released a large amount of OMVs (about $10^9$ particles per milliliter) in crude culture (Nanoparticle tracking analysis, NTA method) (Fig. 1d). As a result, ultracentrifugation of ~25 ml Su_YN1 crude culture led to conspicuous OMV pellets. These serum-induced OMVs showed light-yellow semi-translucent appearance (Fig. 1d). The OMVs in pellets were further confirmed by TEM analysis. FBS-induced OMVs showed more plump morphology than rare OMVs without FBS induction (Fig. 1e). We further quantified OMVs using the Bicinchoninic acid (BCA) assay (total protein quantification) (Fig. 1f), 3-Deoxy-d-manno-2-octulosonic acid (Kdo) assay (lipopolysaccharide quantification), and total lipid quantification (Supplementary Fig. 2g, h). Consistently, these different methods yielded similar results and we quantified OMVs using the BCA assay in further studies.

Bacterias extracellular vesicles are classified according to their types and origins. In addition to OMVs released from living bacterial outer membrane, cell death will lead to release of EOMVs (explosive outer-membrane vesicles) and self-assembled OIMVs (outer-inner membrane vesicles)[18]. We further verified that Su_YN1 OMVs were released by living bacteria as our TEM observation clearly showed that Su_YN1 OMVs originate from the outer membrane (Fig. 1b, lower panel). Moreover, Su_YN1 exhibited robust proliferation both in vivo and in vitro (Supplementary Fig. 3a, b), with even better growth in the presence of serum. The culture of Su_YN1 in the presence of serum also maintained a high level of viability (Supplementary Fig. 3c) and continuously released OMVs under serum stimulation (Supplementary Fig. 3d).

### AmLip is first secreted via T1SS and then loaded onto the OMVs

We next assayed AmLip expression and secretion and found that *AmLip* transcription was not significantly affected by serum (Supplementary Fig. 4a), while AmLip protein expression and secretion was strongly induced by serum (Fig. 2a, Supplementary Fig. 4b). However,

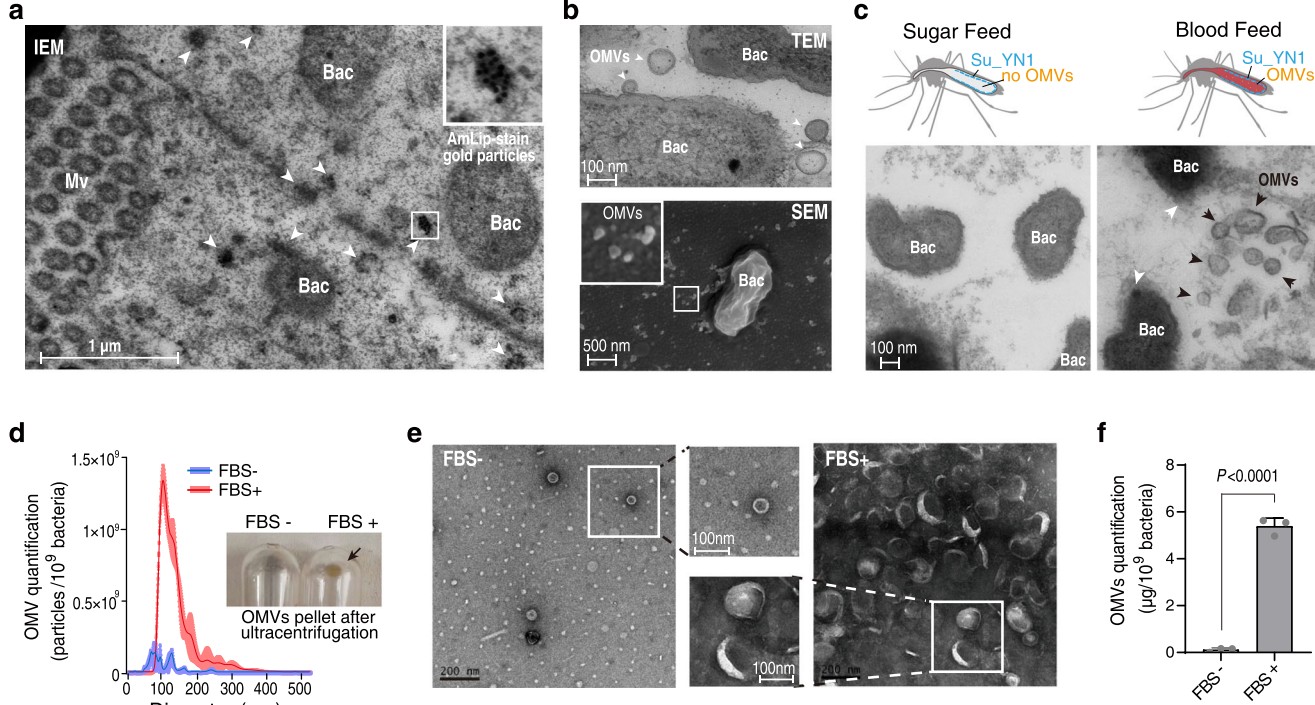

**Fig. 1 | Su_YN1 releases OMVs in the midgut lumen of blood-fed *An. stephensi* mosquitoes. a** Detection of the lipase AmLip by immune electron microscopy (IEM) in an ultrathin cryosection of a midgut of Su_YN1-carrying *Anopheles* mosquito, 24 h after blood feeding. Gold particles indicating AmLip staining are marked by white arrowheads. Mv mosquito microvilli, Bac *Serratia* Su_YN1 bacterium, Scale bar, 1 μm. Similar results were obtained from two biological repeats. **b** Transmission Electron Microscope (TEM, upper panel) and Scanning Electron Microscope (SEM, lower panel) images of *Serratia* Su_YN1 bacterium (Bac) from the mosquito midgut lumen. OMVs produced by Su_YN1 bacterium are indicated by white arrowheads. Scale bar, 100 nm (upper panel) and 500 nm (lower panel). Similar results were obtained from two biological repeats. **c** Comparison of midgut sections by TEM analysis of Su_YN1-carrying *Anopheles* mosquitoes which had fed on either a blood meal or a sugar meal. Black arrowheads, OMVs; white arrowheads, OMVs budding from the surface of *Serratia* Su_YN1 bacteria (Bac). Scale bar, 100 nm. Similar results were obtained from two biological repeats. **d** Nanoparticle

Tracking Analysis (NTA) of Su_YN1 bacterium cultured in RPMI 1640 medium supplemented with or without pre-cleared Fetal Bovine Serum (FBS). OMV concentrations are normalized to $1 \times 10^9$ bacteria. The distribution of particle sizes was demonstrated as polylines (mean ± SD, $n = 3$ independent measurements for each group). Similar results were obtained from two biological repeats. The image in the right panel indicates OMVs pellets in the centrifuge tube bottom (black arrow) after ultracentrifugation. **e** TEM analysis of OMVs ultracentrifugation pellets (negative stained with uranyl acetate) from **d**, Square frames show enlarged images to display Su_YN1-secreted OMVs with or without FBS induction. Scale bar, 100 nm. **f** OMV quantification by bicinchoninic acid (BCA) assay of Su_YN1 cultured with or without FBS (mean ± SD, $n = 3$). OMV concentrations are normalized to $1 \times 10^9$ bacteria. Statistical significance was determined using a one-way ANOVA test, $P$ values are indicated above the plots. Similar results were obtained from three biological repeats. Source data are provided as a Source Data file.

deletion of *AmLip* had no effect on Su_YN1 OMV production (Supplementary Fig. 4c), indicating that AmLip secretion and OMV biogenesis are independently regulated.

Bacterial OMVs are an important vehicle for transporting various molecule cargoes[19]. To confirm the association of AmLip with OMVs, we cultured Su_YN1 in vitro and purified Su_YN1 OMVs by ultracentrifugation[20], followed by proteomic analysis of the OMVs. As expected, the purified OMVs contained high abundance of AmLip (Supplementary data 1). We also used Western blot to detect the distribution of AmLip (Supplementary Fig. 4d), and found that majority of the secreted AmLip protein was associated with OMVs (Fig. 2b), consistent with previous IEM observation of in vivo gut sections. As a result, most of AmLip protein in the supernatant was retained by a 100 KDa membrane filter, larger than AmLip's molecular weight (~64 KDa) (Supplementary Fig. 4e), indicating that this molecule is physically associated with OMVs. We further confirmed this observation by using both IEM (Fig. 2c) and an immunofluorescence assay (Supplementary Fig. 4f). Together, these results reveal that AmLip is associated with OMVs.

Since lipase is normally secreted via a type I secretion system (T1SS)[21,22], we reasoned that AmLip may bind to OMVs after being secreted via T1SS. To test this hypothesis, we generated the AmLip Δ351-614 construct by deleting the C-terminal T1SS-recognition region

of AmLip (Amino sequence 351-614), and then detected the distribution of both AmLip and AmLip Δ351-614 using Western blotting (Supplementary Fig. 5a). As expected, AmLip Δ351-614 retained in the cytosol and was absent from secretion and OMV loading (Supplementary Fig. 5b). To further examine whether AmLip loads OMVs surface or lumen, we performed protease digestion test with trypsin and proteinase K[23,24]. Protease digestion of OMVs led to complete degradation of AmLip, indicating that AmLip is selectively bound to the surface of OMVs after its secretion (Fig. 2d).

### AmLip-loaded OMVs strongly inhibit *Plasmodium* parasites

AmLip is a member of the lipase family (E.C.3.1.1.3) that exhibits strong affinity for oil-water interface, suggesting that the surface of serum-induced Su_YN1 OMVs may favor lipase binding. To determine whether the AmLip-OMV association is specific and dependent on the AmLip protein structure, we expressed and purified full-length AmLip and AmLip fragments. Fragment 1 lacked the C-terminal (Δ351-614) region required for recognition by T1SS, fragment 2 lacked the N-terminal lid structure (Δ2-74), and fragment 3 lacked both the C-terminal and N-terminal structures (Δ2-74; 351-614) (Supplementary Fig. 6a, b). Only the full-length AmLip showed lipase activity on an egg yolk plate (Supplementary Fig. 6c). We incubated these fragments with Su_YN1 OMVs, purified the OMVs, and then detected the binding of these

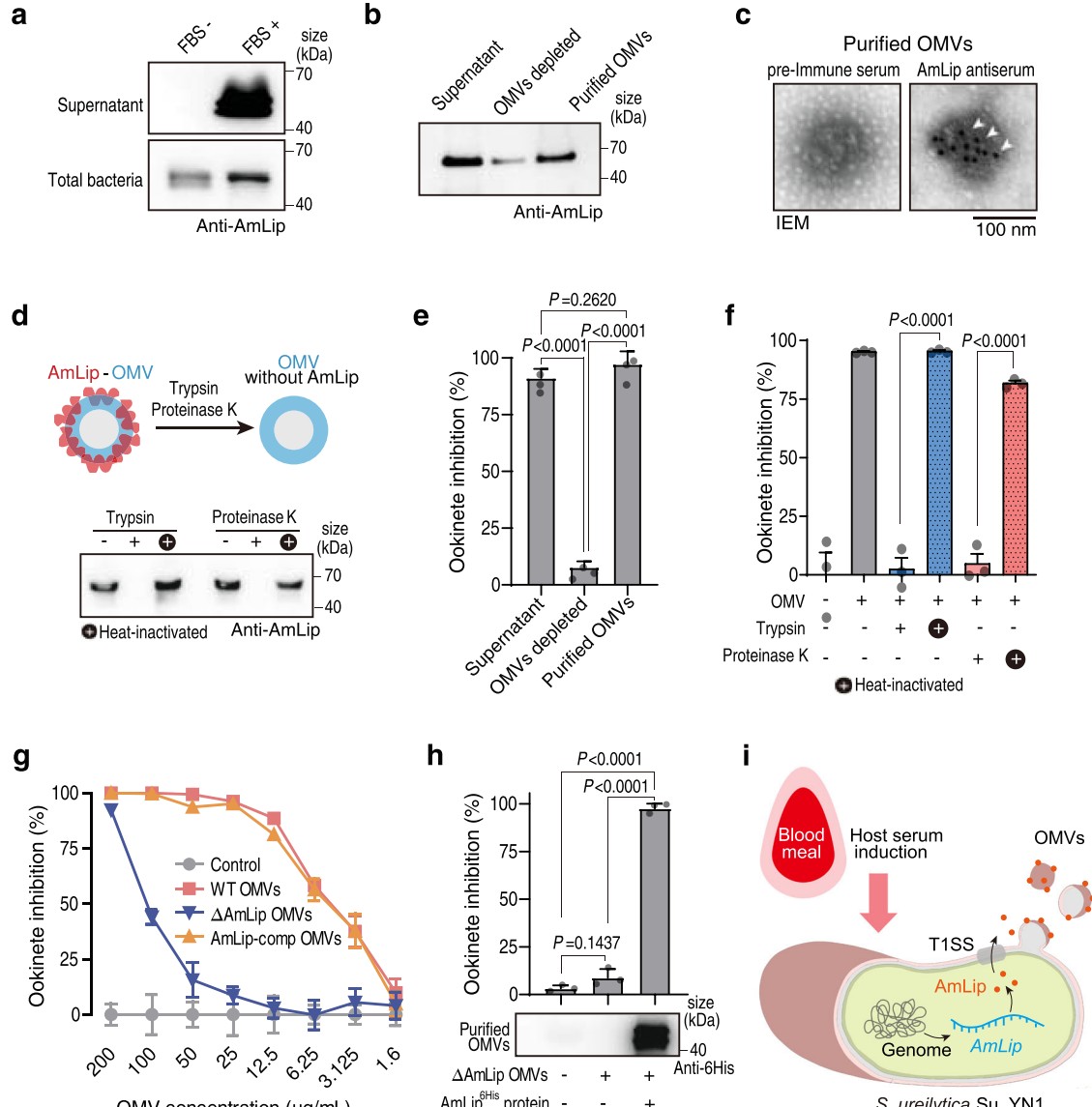

**Fig. 2 | Host serum induces the formation of AmLip-OMV complex that kills *Plasmodium* parasites. a** Western blot detection of AmLip protein secretion into the supernatant of Su_YN1 bacterium cultured with or without FBS. Similar results were obtained from two biological repeats. **b** Western blot detection of AmLip protein in Su_YN1 culture supernatant, OMV-depleted supernatant, and OMVs purified from the culture supernatant using AmLip antiserum. Similar results were obtained from two biological repeats. **c** Representative images of IEM detection of AmLip on Su_YN1 OMVs using pre-immune serum and AmLip antiserum. White arrowheads point to gold particles indicating AmLip staining. Scale bar, 100 nm. Similar results were obtained from two biological repeats. **d** Detection of AmLip on the surface of OMVs. Western blot using AmLip antiserum of OMVs treated (or not) with trypsin or proteinase K. Similar results were obtained from two biological repeats. **e** *P. berghei* ANKA ookinete inhibition assay of Su_YN1 culture supernatant, OMV-depleted supernatant, and purified OMVs resolved in RPMI 1640 (mean ± SD, $n = 3$). Statistical significance was determined using one-way ANOVA test. Similar results were obtained from two biological repeats. **f** *P. berghei* ANKA ookinete inhibition assay by Su_YN1 OMVs treated (or not) with trypsin or proteinase K (mean ± SD, $n = 3$). Statistical significance was determined using a two-tailed Student's *t*-test. Similar results were obtained from two biological repeats. **g** Concentration curve of *P. berghei* ANKA ookinete inhibition by OMVs from wildtype (WT), *AmLip*-KO and *AmLip*-complemented Su_YN1 bacterial culture (mean ± SD, $n = 3$). Statistical details are provided in Source Data file. **h** *P. berghei* ANKA ookinete inhibition assay of OMVs purified from *AmLip*-KO Su_YN1 culture supernatant supplemented with or without 6×His-tagged recombinant AmLip protein (mean ± SD, $n = 3$). Recombinant AmLip protein loading was detected by Western blot (lower panel). Statistical significance was determined using one-way ANOVA test. Similar results were obtained from two biological repeats. **i** Schematic diagram showing host serum induces Su_YN1 to produce OMVs and secrete AmLip, which form AmLip-OMV complex that kills *Plasmodium*. *P* values in **e**, **f** and **h** are indicated above the plots. Source data are provided as a Source Data file.

AmLip fragments using an anti-6His antibody. We found that only the full-length AmLip binds OMVs (Supplementary Fig. 6d), indicating that an intact protein structure is required for both lipase activity and OMV binding.

As the majority of secreted AmLip proteins are associated with OMVs, we next asked whether this association correlates with the anti-*Plasmodium* activity of Su_YN1 culture. We tested anti-*Plasmodium* activity of Su_YN1 culture supernatant, OMVs-depleted supernatant

and purified OMV fraction on *Plasmodium* ookinetes. Anti-*Plasmodium* activity was largely associated with the OMVs fraction (Fig. 2e) and protease digestion of OMVs abolished anti-*Plasmodium* activity (Fig. 2f). To further test whether AmLip binding confers OMVs anti-*Plasmodium* activity, we knocked out the bacterium *AmLip* gene and compared anti-*Plasmodium* activity of OMVs from the wild-type Su_YN1, its *AmLip*-knockout (*AmLip*-KO) mutant and the *AmLip*-complemented strains. *AmLip*-KO OMVs were severely deficient for anti-

*Plasmodium* activity, while OMVs from the *AmLip*-complemented strain recovered antimalarial activity (Fig. 2g). These results indicate that OMVs exert anti-malarial activity through transport of the anti-*Plasmodium* lipase AmLip to the parasites.

To further demonstrate that AmLip could bind to the OMVs, and its binding endows OMV anti-*Plasmodium* activity, we added recombinant 6×His-tagged AmLip protein to a Su_YN1 *AmLip*-KO culture and then purified the OMVs. Western blot analysis indicated efficient loading of recombinant AmLip protein onto the OMVs (Fig. 2h). As expected, loading of recombinant AmLip protein restored the anti-*Plasmodium* activity of *AmLip*-KO OMVs (Fig. 2h). Collectively, these results indicate that host serum simultaneously induces OMVs production and AmLip secretion, leading to the formation of AmLip-OMV complex with potent *Plasmodium*-killing activity (Fig. 2i).

## Su_YN1 OMVs inhibit *Plasmodium* infection in the mosquito gut

We next asked whether Su_YN1-released OMVs indeed block *Plasmodium* infection in the mosquito gut under physiological conditions. We introduced the Su_YN1 wildtype strain and AmLip Δ351-614 strain (that could not secrete AmLip to form AmLip-OMVs complex) into *An. stephensi*. Although these strains showed comparable gut colonization

(Supplementary Fig. 7a), AmLip Δ351-614 strain lost the anti-*Plasmodium* activity (Supplementary Fig. 7b), indicating that AmLip secretion and loading onto OMVs are critical for Su_YN1 anti-*Plasmodium* activity in vivo.

To directly show that AmLip-loaded OMVs inhibit *Plasmodium* in the mosquito gut, the purified OMVs were added to the infectious blood and fed to *An. stephensi* using a Standard Membrane Feeding Assays method (Supplementary Fig. 8)[25]. We found that Su_YN1 OMVs strongly inhibited oocyst formation in the mosquito midgut (Fig. 3a). Moreover, the inhibition activity depends on AmLip loading (Supplementary Fig. 9a). We next tested the *Plasmodium* inhibition activity of OMVs under various concentrations, and found that as low as 2 μg/ml OMVs can significantly reduce oocyst load and the prevalence rate, indicating that OMVs are highly potent (Supplementary Fig. 9b).

Next, to show the direct targeting of Su_YN1 OMVs on *Plasmodium* cells, *P. berghei* zygotes and ookinetes were co-cultured for 5 min with purified OMVs and fixed for SEM and TEM analysis. We found that OMV particles bind to the parasite membrane and cause parasite lysis (Fig. 3b, c). Addition of Su_YN1 OMVs to ookinetes rapidly led to ookinete cell damage (mCherry fluorescence disappears) and loss of

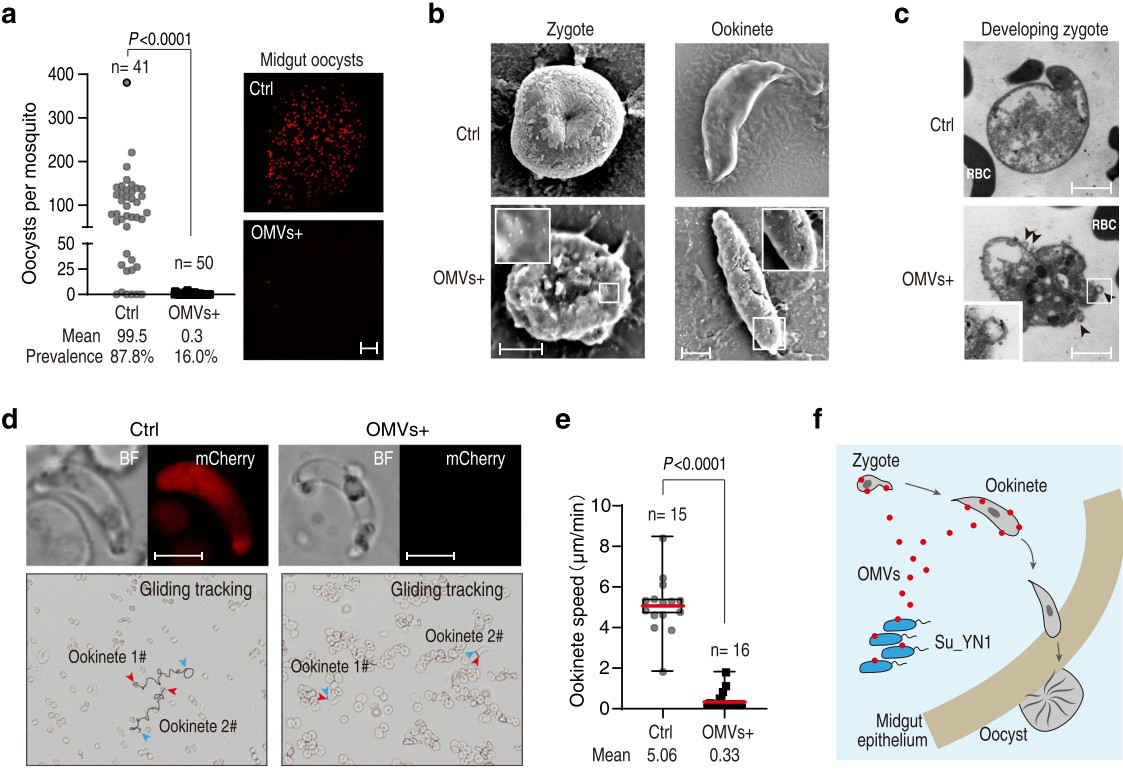

**Fig. 3 | Su_YN1 OMVs target early midgut stages of *Plasmodium* parasites.**
**a** Oocyst load in the midgut of *An. stephensi* mosquitoes fed with a fluorescent mCherry transgenic *P. berghei* ANKA strain infectious blood supplemented with PBS (Ctrl) or OMVs. Oocyst load is shown in the left panel. Right panel shows representative fluorescence microscopy images (mCherry signals indicate oocysts) of *An. stephensi* midguts. Scale bar, 50 μm. Significant difference in oocyst intensities between two groups was analyzed using the two-tailed Mann−Whitney test, *P* values are indicated above the plots. **b** SEM observation of *P. berghei* ANKA zygote and ookinete 5 min after incubation with Su_YN1 OMVs. Scale bar, 5 μm. Similar results were obtained from two biological repeats. **c** TEM images of sections of *P. berghei* ANKA developing zygotes after incubation with or without Su_YN1 OMVs. Scale bar, 5 μm. Similar results were obtained from two biological repeats. **d** Effect of OMVs on ookinete gliding. Live tracking of ookinete gliding on Matrigel matrix was recorded. Ookinetes were pretreated with OMVs at final concentration of 50 μg/ml. Reconstruction of tracked ookinete paths from high-magnification

movies reveal movement patterns. Blue arrowheads and red arrowheads indicate gliding start and stop positions respectively, and the lines indicate the ookinete gliding trajectory. Live ookinetes treated (or not) with OMVs are shown in the upper panels. Ookinetes lost mCherry fluorescence after OMV treatment, indicating damage of cell viability. Scale bar, 5 μm. Similar results were obtained from two biological repeats. **e** Gliding speed measurement of *P. berghei* ANKA ookinetes. Ookinetes were incubated with OMVs or PBS (Ctrl) for 5 min prior to motility tests. The middle line of box-plot diagram represents median, boxes extend from the 25th to 75th percentiles. The whiskers mark the 10th and 90th percentiles. Significant difference in gliding motility between two groups was analyzed using the two-tailed Mann−Whitney test, *P* values are indicated above the plots. Similar results were obtained from two biological repeats. **f** Schematic diagram showing Su_YN1 secreted OMVs targeting developing zygote and ookinete in the mosquito midgut. Source data are provided as a Source Data file.

ookinete gliding motility (Fig. 3d, e, and Supplementary movie 1 and 2). Collectively, these results indicate that AmLip-loaded OMVs secreted by *S. ureilytica* Su_YN1 efficiently bind to and kill *Plasmodium* parasites in the mosquito gut, rendering mosquito refractoriness to *Plasmodium* infection (Fig. 3f).

### OMV-bound AmLip selectively enters and disrupts *Plasmodium* parasites

We next investigated whether Su_YN1 OMVs could also target asexual stage parasites. Human *P. falciparum* 3D7 (Pf3D7) asexual stage parasites were treated with purified OMVs for 5 min and then cultured. OMV treatment strongly inhibited Pf3D7 asexual replication (Fig. 4a), indicating that Su_YN1 OMVs target mosquito midgut stages as well as asexual blood stages of malaria parasites. Notably, we noticed that OMVs can disrupt sexual and asexual *Plasmodium* parasites selectively while leaving uninfected red blood cells (RBC) intact (Fig. 4a).

To directly observe OMV entry into *Plasmodium* parasites, purified OMVs were stained with DiI (red) or DiO (green) as previously described[26]. When *Plasmodium* parasites were co-incubated with the labeled OMVs and observed with a fluorescence microscope, labeled OMV signal could be detected in both sexual and asexual parasites shortly after co-incubation (Supplementary Fig. 10a).

To further observe how rapidly OMVs enter and disrupt the parasites, we used asexual stages of *P. falciparum* 3D7 as these parasites are standardized for in vitro culture, immobile, and suitable for living tracking. We added the stained OMVs in the Pf3D7 asexual culture and traced OMV fluorescent signal on the living parasites. OMV signal could be detected on parasitized host cell membrane and parasites as soon as one minute after co-incubation (Fig. 4b). OMV accumulation led to rapid parasite membrane destruction and cell death (Supplementary video 3). The parasitized host cell membrane appeared blistered and shrunk, parasites collapsed within several

minutes (Fig. 4b, c), the parasite vacuole was damaged, and the nucleus disintegrated (Supplementary Fig. 11a). Of note, Su_YN1 OMVs selectively enter and kill *Plasmodium* parasites and parasitized RBCs and do not affect healthy RBCs (Supplementary Figs. 10b and 11b). Immunofluorescence assays showed efficient delivery of OMV-bound AmLip to the parasites (Fig. 4d). Although *AmLip*-KO OMVs also efficiently entered *Plasmodium* cells (Supplementary Fig.11c), they did not lyse the parasites (Supplementary video 4). Immuno-fluorescence analysis of OMV-treated parasite cells revealed destruction of the parasitized host membrane and of the *Plasmodium* parasite parasitophorous vacuole membrane (PVM) (Fig. 4e) by wild type OMVs but not *AmLip*-KO OMVs, which further confirmed that OMV-mediated antimalarial activity depends on the loading of lipase AmLip. Collectively, these findings reveal that Su_YN1 OMVs selectively target and rapidly destroy *Plasmodium* parasites through the loaded AmLip, representing a novel mechanism by which gut symbiotic bacteria deliver anti-*Plasmodium* effectors via OMVs to kill malaria parasites.

### Su_YN1 OMVs are enriched with certain serum-derived phospholipids

We next investigated how Su_YN1 OMVs specifically deliver AmLip to the malaria parasite. We harvested OMVs from Su_YN1 cultured with exosome-depleted FBS (FBS+ OMVs) or without FBS (FBS- OMVs) in RPMI 1640 medium. These OMVs were stained with DiI and incubated with asexual *P. falciparum* 3D7 parasites. We found that only FBS+ OMVs but not FBS- OMVs, efficiently enter *Plasmodium* cells (Fig. 5a).

Bacterial OMVs are phospholipid-based nanoparticles that originate from the outer membrane[16]. Moreover, OMV composition is significantly influenced by the environment[27,28]. The FBS+ OMVs released under host serum-stimulating conditions may have unique lipidome. We extracted the total lipid of Su_YN1 FBS+ OMVs and FBS- OMVs, and

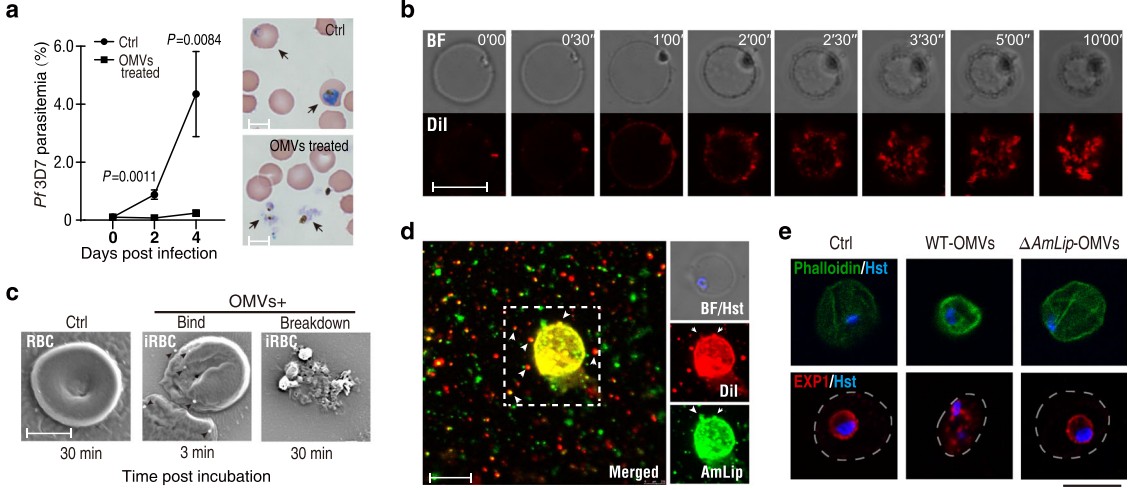

**Fig. 4 | Su_YN1 OMVs efficiently enter *Plasmodium* and kill the parasite. a** Killing effect of OMVs on asexual-stage *P. falciparum* 3D7 parasites. Parasites were pre-treated with OMVs for 5 min at a final concentration of 100 μg/ml. Untreated (Ctrl) and OMV-treated parasites were cultured and parasitemia was monitored (mean ± SD, *n* = 3). Parasite Giemsa staining at day 2 is shown in the right panels. Black arrowheads indicate *Plasmodium* infected RBC (upper panel) and lysed parasites (lower panel). Scale bar, 5 μm. Similar results were obtained from two biological repeats. Statistical significance was determined using a two-tailed Student's *t* test. **b** Confocal live tracking of DiI-labeled OMVs (red) taken up by *P. falciparum* 3D7 asexual parasites. Images were captured at 30 s intervals for 10 min after addition of DiI-labeled OMVs to the parasite culture. BF bright field. Scale bar, 5 μm. Similar results were obtained from two biological repeats. **c** SEM observation of a red blood cell (RBC) and of a *P. falciparum* 3D7-infected RBC (iRBC) incubated with OMVs. Black arrowheads indicate OMVs bound to the iRBC

surface. Scale bar, 5 μm. Similar results were obtained from two biological repeats. **d** Indirect immunofluorescent assay of asexual *P. falciparum* 3D7 parasites co-incubated with DiI-stained Su_YN1 OMVs at final concentration of 50 μg/ml. AmLip immunofluorescent and OMV-DiI signals largely co-localize and strongly accumulate on the parasite. BF, bright field. Parasite nuclei are stained with Hoechst 33342 (Hst). Scale bar, 5 μm. Similar results were obtained from two biological repeats. **e** Indirect immunofluorescence detection of host RBC cytoskeleton using FITC-phalloidin (upper panel) and parasite parasitophorous vacuole membrane (PVM) labeled with EXP1 antiserum (lower panel). *P. falciparum* 3D7 parasites were incubated with OMVs purified from wildtype (WT) and *Amlip*-KO Su_YN1 bacterial culture for 5 min and fixed for detection. Parasite nuclei are stained with Hoechst 33342 (Hst). Scale bar, 5 μm. Similar results were obtained from two biological repeats. Source data are provided as a Source Data file.

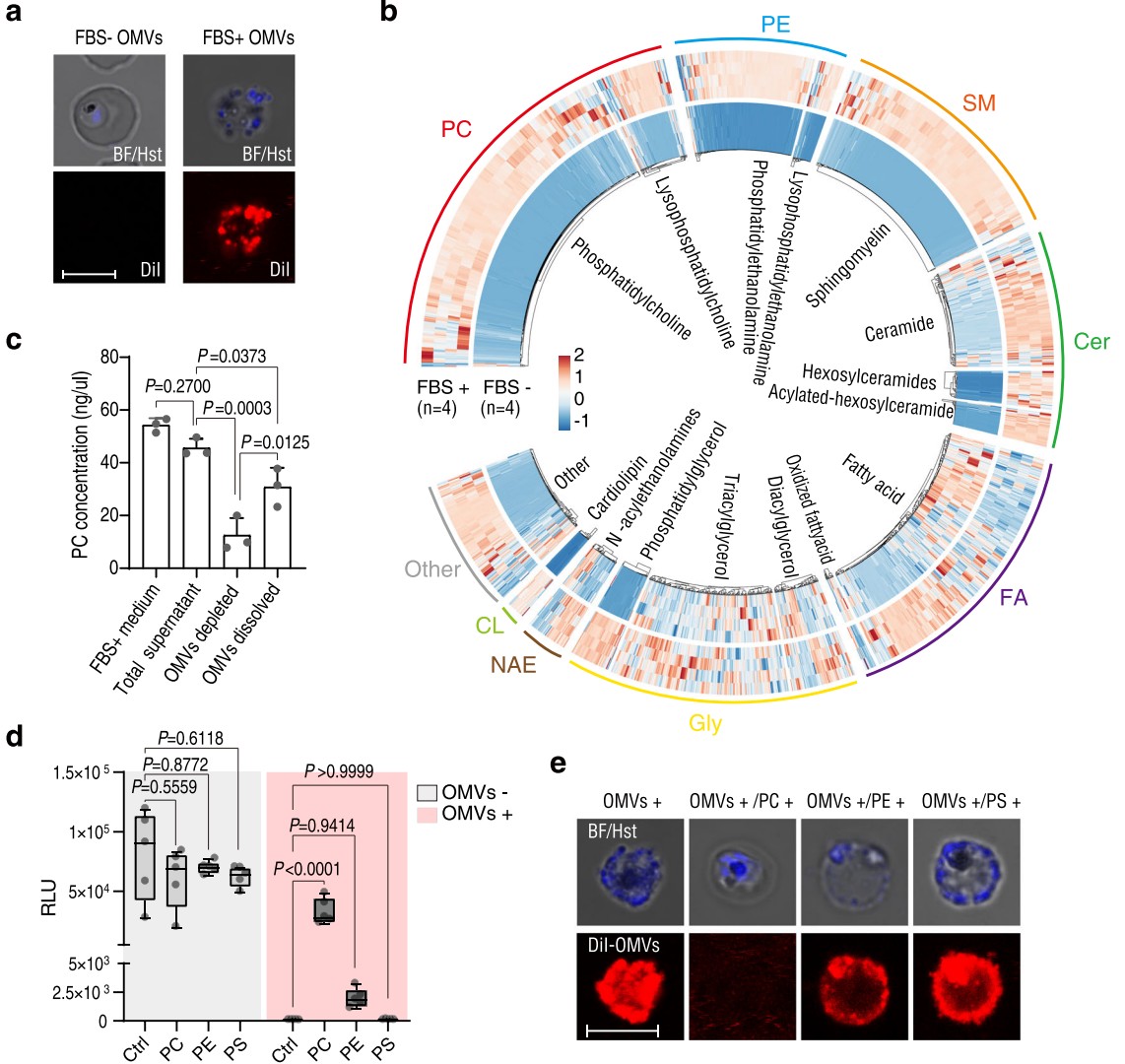

**Fig. 5 | Su_YN1 OMVs incorporate certain serum-derived lipids that mediate OMV entry into the parasite via the *Plasmodium* phosphatidylcholine scavenging pathway. a** Uptake of DiI-stained OMVs from *Serratia* Su_YN1 cultured with FBS (FBS+ OMVs) or without FBS (FBS- OMVs), by *P. falciparum* 3D7 asexual parasites. The parasite nucleus was stained with Hoechst 33342 (Hst). BF, bright field. Scale bar, 5 μm. Similar results were obtained from two biological repeats.
**b** Clustering analysis of lipid contents of FBS+ OMVs and FBS− OMVs. The individual rows indicate lipids detected and the individual columns indicate independent samples (mean ± SD, n = 4 independent repeats in each group). Specific enriched lipid clusters in FBS+ OMVs are delimited with a square frame.
**c** Phosphatidylcholine (PC) quantification of the RPMI 1640 plus 10% FBS medium (FBS+ medium), Su_YN1 supernatant cultured in FBS+ medium (Total supernatant), OMVs depleted total supernatant (OMVs depleted), and OMVs fraction (OMVs dissolved), to evaluate the distribution of PC (mean ± SD, n = 3 independent repeats in each group). Statistical significance was determined using one-way ANOVA test.

Similar results were obtained from two biological repeats. **d** Effect of different phospholipids on inhibition of *Plasmodium* by OMVs. *P. berghei* ANKA luciferase asexual stage parasites were cultured in dishes pre-coated with phospholipids, prior to adding Su_YN1 OMVs. Parasite viability was estimated by measuring luciferase relative light units (RLU) (n = 4). The middle line of box-plot diagram represents median, boxes extend from the 25th to 75th percentiles. The whiskers mark the 10th and 90th percentiles. Similar results were obtained from two biological repeats. Statistical significance was determined using one-way ANOVA test, *P* values are indicated above the plots. **e** Effects of PC on antagonizing OMV uptake by *P. falciparum* 3D7 asexual stage parasites, which were cultured in dishes pre-coated with phospholipids, then incubated with DiI-stained Su_YN1 OMVs. Uptake of OMVs was estimated by observing DiI fluorescent signals on parasites. The parasite nucleus was stained with Hst. BF, bright field. Scale bar, 5 μm. Similar results were obtained from two biological repeats. Source data are provided as a Source Data file.

conducted lipidome analysis. The lipidome composition of FBS+ OMVs was distinct from that of the FBS- OMVs. In particular, FBS+ OMVs contain substantial amounts of major phospholipids including phosphatidylcholine (PC), phosphatidylethanolamine (PE) and sphingomyelin (SM) (Fig. 5b, Supplementary Data 2).

We further asked whether the lipidome of Su_YN1 FBS+ OMVs is unique among bacterial OMVs, and whether this distinctiveness is responsible for *Plasmodium* targeting. We purified OMVs from *Escherichia coli* K12 (K12 OMVs) cultured in Luria Broth (LB) medium as *E. coli* K12 could not grow in RPMI1640 medium and survive in FBS. As a

comparison, OMVs from Su_YN1 cultured in LB (LB OMVs) were also prepared. Both K12 OMVs and LB OMVs failed to enter *Plasmodium* cells (Supplementary Fig. 12a). Comparative lipidome analysis further revealed that the FBS+ OMV lipid composition is distinct from the other OMVs (Supplementary Fig. 13a, b) and that the FBS+ OMVs enriched with significantly higher levels of certain lipids – especially major phospholipids including PC, PE, SM – as compared to FBS- OMVs, LB OMVs or K12 OMVs (Supplementary Fig. 13b and Supplementary Data 3). Collectively, the FBS+ OMVs released by Su_YN1 under serum induction harbor specific lipids characterized by substantial PC, PE and SM.

Bacterial membranes are predominantly composed of lipids including phosphatidylglycerol (PG), PE and cardiolipin (CL)[28]. Although PC is the major membrane-forming phospholipid in eukaryotes, only a small portion of the domain bacteria are estimated to have PC in their membrane[29]. Notably, PC and PE are the most abundant phospholipids in serum[30]. Our results indicate that Su_YN1 may efficiently incorporate host serum-derived lipids especially PC into the outer membrane to produce specialized OMVs. To further confirm incorporation of serum PC into Su_YN1 OMV biogenesis, we quantified the PC levels of Su_YN1 bacteria and culture supernatants. PC was hardly detected in Su_YN1 bacterial cells when cultured in medium without FBS (FBS-), and PC were detected in Su_YN1 cells when cultured in serum containing medium (Supplementary Fig. 12b). However, the PC level in the OMVs-depleted supernatant was decreased, and high level of PC was detected in the purified OMVs, indicating uptake of serum PC by the bacteria and integration of PC into OMVs (Fig. 5c). Moreover, FBS+ OMVs contained a significantly higher PC content than in FBS- OMVs, LB OMVs or K12 OMVs (Supplementary Fig. 12c). To directly demonstrate PC incorporation during Su_YN1 OMV biogenesis, we added NBD-fluorescently labeled PC (NBD-PC) into Su_YN1 culture and purified the OMVs. Fluorescence nanoparticle analysis indicated a high level of incorporation of NBD-PC into OMVs (Supplementary Fig. 13c). NBD fluorescence was detected on both Su_YN1 bacteria and its OMVs, with the majority of the NBD fluorescence detected in the OMV fraction (Supplementary Fig. 12d, e).

### *Plasmodium* phosphatidylcholine scavenging pathway mediates OMV uptake

Malaria parasites lost several lipid metabolic pathways during evolution and must scavenge certain lipids from the host[31,32]. Host phospholipids, lipoate and cholesterol for example, are key mediators of malaria parasite survival[33,34]. The presence of serum-derived phospholipids may make Su_YN1 FBS+ OMVs ideal substrates for uptake by the parasites. We coated the bottom of culture dishes with different phospholipid components and investigated whether any of them antagonizes the uptake of OMVs by the luciferase-expressing asexual-stage parasites (Supplementary Fig. 14a). We chose PC and PE, the two highly enriched phospholipids in FBS+ OMVs as well as in host serum, for the antagonism test. It was not possible to conduct the antagonistic test with SM because of its poor solubility and fact that SM and its derivative ceramides are reported to have anti-*Plasmodium* activity[35,36]. This may explain the *AmLip*-KO OMV's anti-malarial activity at very high concentrations (Fig. 2g). We also tested phosphatidylserine (PS) and found no obvious antagonistic effect on the OMV's anti-*Plasmodium* activity. However, addition of PC and PE significantly antagonized OMV anti-*Plasmodium* activity, as indicated by parasite luciferase relative light units (RLU) (Fig. 5d). Moreover, PC exhibited concentration-dependent antagonistic effect on OMV's anti-*Plasmodium* activity (Supplementary Fig. 14b). Fluorescence microscopic analysis confirmed that PC pretreatment strongly inhibited OMV uptake by Pf3D7 asexual stage parasites (Fig. 5e). Likewise, addition of PC also strongly antagonized the killing effect of Su_YN1 OMVs against *Plasmodium* ookinetes (Supplementary Fig. 14c).

*Plasmodium* constantly takes up host PC, which is quickly translocated from host cell membrane to the PVM[34]. Notably, host membrane-derived PC is highly favored by the parasites, which does not undergo appreciable remodeling following uptake and is used directly in the biogenesis of the parasite membranes[34]. Our observation of the rapid intake of Su_YN1 OMVs and quick translocation of OMVs from host membrane to the parasite membrane indicates that OMVs may enter the parasites through the PC scavenging pathway. To investigate the involvement of the PC scavenging pathway in OMV uptake, we pretreated Pf3D7 asexual parasites with NBD-PC prior to incubation with DiI-stained OMVs, and monitored OMV uptake by the parasites using flow cytometry. NBD-PC rapidly entered the parasites

(Supplementary Fig. 15a) and significantly reduced OMV uptake (Supplementary Fig. 14d, e). *Plasmodium* PC scavenging stalls at 4 °C, is highly active at 37 °C, and can be inhibited by ATP depletion[37,38] (Supplementary Fig. 15a). We compared OMV uptake at 4 °C or 37 °C and observed little OMV uptake at 4 °C, while uptake was quickly activated at 37 °C (Supplementary Fig. 15b). Moreover, depletion of parasite ATP by sodium azide treatment significantly inhibited OMV uptake by the parasites (Supplementary Fig. 15b). To more specifically demonstrate OMV entry into the parasites via PC scavenging, we treated the parasites using L-α Glycerophosphorylcholine (L-α GPC), which contains same choline head group structure as PC and antagonizes PC uptake[37,38]. L-α GPC treatment significantly inhibited PC (Supplementary Fig. 15c) and OMV (Supplementary Fig. 15d) uptake by the parasites. Collectively, these findings indicate that the PC scavenging pathway mediates the uptake of Su_YN1 OMVs by *Plasmodium* parasites.

## Discussion

Releasing membrane vesicles by bacteria is an important intercellular communication mechanism. Most functional studies of bacterial OMVs have been focused on mammalian-bacterial interactions. OMVs released from pathogenic bacteria play essential roles in pathogen-host interactions including the transmission of virulence factors into host cells and modulation of host defense[19,39]. In this study we report that OMVs released from a mosquito gut symbiotic bacterium after a blood meal play an important role in defense against gut pathogens by delivering an effector protein to *Plasmodium* parasites (Supplementary Fig. 16). The discovery of these inter-kingdom interactions between a gut microbe and a eukaryotic parasite reveals a new paradigm by which a commensal bacterium of the mosquito microbiota selectively delivers the effector molecule to a gut parasite.

Bacterial OMV production is induced by a variety of conditions such as growth stage and nutrition, temperature and oxidative stress[18]. Like most other hematophagous invertebrates, mosquitoes require a blood meal to produce eggs. Our study reveals that blood serum from the mammalian host serves as a novel potent stressor to stimulate OMV production by the gut symbiotic bacterium. Host serum contains substantial lipids, which may be diverted by gut bacteria[40] and promote their lipid metabolic reprogramming, leading to changed membrane homeostasis[41]. Moreover, digestion of vertebrate blood results in the release of excess heme and iron derived from degradation of hemoglobin and the serum iron transport protein transferrin[42,43], which can promote oxidative challenge to gut microbiota[44]. Membrane instability is a common feature for excessive OMV generation[45]. Lipid metabolic reprogramming and oxidative stress may disturb the stability of phospholipid bilayers, which may lead to membrane budding and OMV release.

OMVs are produced in gram-negative bacteria by blebbing of the outer membrane. OMV composition is generally similar to the parent bacterium's membrane, but OMV composition and contents are variable depending on the bacterial growth stage and environmental conditions[28]. There is evidence of preferential packaging of proteins, lipids, nucleic acids, and polysaccharide into OMVs[46], although the sorting mechanisms to direct the components to OMVs remains elusive[47]. We found that host serum induction reshapes Su_YN1 OMV's lipid composition with the incorporation of serum-derived lipids. Notably, phospholipids PC and PE, which are abundant in host serum, are highly enriched in Su_YN1 OMVs cultured in serum-containing medium, and this facilitates specific OMV-*Plasmodium* parasite interaction. Interestingly, direct application of PC to the Su_YN1 culture did not significantly induce OMV biogenesis (Supplementary Fig. 13d), suggesting that lipoproteins are likely the source of PC utilized by the bacteria, and the protein portion may be important for the uptake of PC. Alternatively, other factors present in serum may be responsible for inducing robust OMV biogenesis.

In this study, we found that Su_YN1 OMVs selectively target the *Plasmodium* parasites (both mosquito midgut and asexual blood stages). Host derived lipids are essential for protozoan parasite survival[48]. Eukaryotic parasites such as *Toxoplasma* spp and *Trypanosoma* spp also ingest substantial phospholipids. It was reported that *T. gondii* has synthetic capacity only to produce ~5%–10% of phosphatidylcholine (PC) as required for this parasite doubling[49], and *T. gondii* also actively salvage PS and PE[50]. Although most eukaryotic parasites retain some of the essential phospholipid synthetic pathways (for example, the Kennedy pathway for PC and PE synthesis), they still need to salvage substantial phospholipids especially at the active proliferative stages[32,51]. Thus, entry of membrane vesicles containing specific lipids into eukaryotic parasites through lipid salvage is likely a general phenomenon yet to be identified in other microbe-pathogen interactions.

Malaria parasites actively internalize phospholipids from its erythrocyte membrane and the extracellular medium[52], although the exact transporters and subsequent pathways responsible for specific phospholipids intake remains ambiguous[53]. Previous studies showed that vesicles are key mediator of cargo transport for malaria parasites[54]. The high enrichment of serum-derived lipids and their proper size range may promote Su_YN1 OMV interactions with parasites. Our findings indicate that the PC scavenging pathway plays a crucial role in mediating OMV uptake by malaria parasites, although other lipid uptake pathways or membrane receptors may also be involved. The active PC scavenging of parasitized RBCs distinct from healthy RBCs, which do not uptake OMVs and are exempt from being lysed by the OMV cargo AmLip. Notably, *Plasmodium* parasites take up OMVs via the PC scavenging pathway rapidly, which is different from the previously reported LPS-mediated entry[55] and lipid-raft mediated intake pathways[56]. These findings indicate that eukaryotic parasites have evolved a novel vesicle uptake mechanism to meet their unique nutritional demands.

OMVs play a variety of functional roles yet the OMV's ultimate function is determined by their unique cargo[47]. Growing evidences indicate that OMV cargo selection is a regulated process, however mechanisms of how cargo proteins are selectively packaged into OMVs remain largely elusive[46,57,58]. The antimalarial protein AmLip is secreted via T1SS to extracellular spaces before binding onto OMV's surface. The unique hydrophobic properties of lipase may explain the affinity of AmLip to OMVs[59,60]. We found that the intact AmLip protein with lipase activity is preferentially accumulated on the surface of *Serratia* Su_YN1 OMVs, indicating that Su_YN1 OMVs possess a unique and specific affinity for binding to its own secreted lipase. Overall, our results provide a subtle regulation in which specialized OMVs assist a T1SS secreted effector protein in targeted cross-kingdom interaction.

In the paratransgenic strategy, T1SS was employed to drive gut symbionts of host insects to secret effector proteins that interfere with pathogen transmission[7,10,13,61]. However, T1SS-secreted effector proteins are not directly transported to target cells and not protected after secretion. Incorporation into OMVs can impart a variety of beneficial properties to protein cargoes, including protection from proteolytic degradation[62], enhancement of long-distance transport[63], as well as specificity in recipient cell targeting as shown here. The interaction between the lipase AmLip with OMVs is necessary for potent anti-*Plasmodium* activity. When AmLip is not associated with OMVs, it diffuses after release and only weakly targets *Plasmodium* cells. However, binding to OMVs strongly enhances the malaria targeting of AmLip, leading to a robust anti-*Plasmodium* activity. Notably, *Serratia* Su_YN1 can stably colonize and proliferate in the mosquito midgut, thereby the delivery of OMVs into *Plasmodium* parasites is in a cumulative manner during parasite development in the midgut. Delivery of anti-*Plasmodium* effector molecules by OMVs to the *Plasmodium* parasite membrane and the infected RBC (iRBC) membrane (which is significantly reorganized by the parasites) promotes accumulation of AmLip protein on these membranes, leading to the effector AmLip-mediated degradation of the membrane structure, and swelling and degradation of the iRBC membrane and parasite disintegration, respectively. In summary, Su_YN1 OMVs represent an ideal effector delivery system that selectively and precisely targets malaria parasites, providing a novel strategy for the development of intervention tools to fight this deadly disease.

## Methods

### Ethics statement
This study was carried out in accordance with the guidelines of the CAS Center for Excellence in Molecular Plant Sciences (Shanghai Institute of Plant Physiology and Ecology) Animal Care and Use Committee (A01MP2001). Six-week-old male SPF (specific pathogen free) grade ICR (Institute of Cancer Research) mice were used for *Plasmodium berghei* infection and infectious blood collection. O-type human blood was provided by Shanghai Red Cross Blood Center (Approval number: shblood2019-28) with signed informed consent form.

### Transmission electron microscopy (TEM) analysis
For midgut samples: The midgut sections were embedded in Epson resin for sectioning. The resulting ultrathin samples were then treated with 2% uranium acetate and lead citrate prior to TEM (model no. H-7700, Hitachi) analysis.

For OMV samples: 3 µl of the OMV preparations were applied onto carbon-coated copper grids (Cat# BZ10021b, ZJKY) and allowed to absorb for 3 min. Then the excess liquid was discarded, and the samples were negatively stained with 2% (wt/vol) uranyl acetate for 3 min and evaluated by TEM.

For *Plasmodium* samples: Parasites were washed with PBS and centrifuged at 1000 g for 3 min. The resulting cell pellets were fixed in 2.5 % glutaraldehyde and subsequently sectioned for TEM analysis.

### Scanning electron microscope (SEM) analysis
For bacteria samples: bacterial suspension was applied to cover slips and then fixed in 2.5% glutaraldehyde (3802-75 ML, Sigma-Aldrich) for 1 h. Subsequently, the cover slips were washed four times with PBS buffer. Samples were then dehydrated in a series of increasing concentrations of ethanol (10% steps) for 10 min each. After critical-point drying, samples were sputter-coated with a thin (10-nm) layer of gold and stored dehydrated until analyzed in the Field Emission Scanning Electron Microscopy (Carl Zeiss, Oberkochen, Germany).

For *Plasmodium* samples with OMV binding: *Plasmodium* parasites treated with OMVs were washed with PBS, then smeared on a cover slip, air dried for 5 min, and then fixed with 100% methanol. After air drying, the cover slip was immediately sputter-coated with gold and analyzed using Field Emission Scanning Electron Microscopy.

### Immune electron microscopy (IEM) analysis
Sectioned mosquito gut samples on carbon-coated copper grids, or OMV-adsorbed carbon-coated copper grids were fixed in 2% (w/v) fresh-made paraformaldehyde and 0.5% (v/v) glutaraldehyde in 0.1 M of PBS buffer at pH 7.4. Samples were blocked with 5% (w/v) of bovine serum albumin (A23088-100G, Abcone) in PBS containing 0.1% (v/v) TWEEN 20 for 1 h. Samples were incubated with AmLip antiserum prepared as previous described[11] or negative serum (1:10), CD63 antibody (A22343, ABclonal) (diluted 1:10), and CD9 antibody (Santa Cruz Cat#sc-59140) (diluted 1:10) for 2 h, and rinsed in TBS-TWEEN 20 buffer three times. The samples were incubated with secondary antibodies (1:50) conjugated to 10-nm gold particles (Goat Anti-Mouse IgG H&L Gold, Cat# bs-0296G-Gold, BIOSS) for 1 h, and also rinsed in TBS-TWEEN 20 buffer three times. Sections were imaged at 80 KV with a Transmission Electron Microscope (model no. H-7700, Hitachi).

## Bacteria culture and preparation of OMVs

In general, OMVs were isolated following a procedure with minor adaptations[20]. Bacteria were cultured overnight in RPMI 1640 (10-041-CV, Corning) with or without 10% FBS (10099141C, Thermofisher) at 30 °C, unless stated otherwise. The FBS was pre-cleared by ultracentrifugation (250,000 g, 4 °C, 3 h) to remove native vesicles and filtered through a 0.22 µm filter (SLGPR33RB, Merck) before use. Bacterial cultures were centrifuged at 4,000 rpm for 20 min, the supernatants were sequentially filtered through a 0.45 µm filter (SLHP033RS, Merck) and a 0.22 µm filter (SLGPR33RB, Merck). The OMVs in the filtered supernatant were pelleted through subsequent ultracentrifugation (250,000 g, 4 °C, 1 h) in 26 ml ultracentrifuge tubes (355654, Beckman) using a Backman 70Ti rotor. OMV pellets were washed in PBS and ultracentrifuge again. Washed OMV pellets were resuspend in 1.5 ml PBS (for quantification and biochemistry tests) or RPMI 1640 medium (for *Plasmodium* culture tests).

## OMV quantification analysis

OMVs were quantified using following three different methods.

BCA assay: The concentration of OMV protein was determined using BCA Protein Assay (Pierce™ BCA Protein Assay Kit, 23227) according to the manufacturer's manual and normalized to $10^9$ bacteria of the respective culture.

KDO assay: Briefly OMV 2-keto-3-deoxyoctonate (Kdo) content was determined as described previously[1] by using Kdo ammonium salt (k2755, Sigma, St. Louis, MO) as a standard sample and normalized to $10^9$ bacteria of the respective culture.

Total lipad assay: OMV lipid content was quantified by using lipid quantification kit (STA-613, Cell Biolabs) according to the manufacturer's manual and normalized to $10^9$ bacteria of the respective culture.

## OMV protease digestion assay

Protease digestion tests of OMV surface proteins were carried out as described previously[2]. OMVs (500 µg/ml in the reaction) were digested using Trypsin (T8150, solarbio) (37 °C, pH 7.4, final concentration of 100 µg/ml) or Protease K (T8936, TargetMol) (50 °C, pH 7.4, final concentration of 100 µg/ml) for 30 min and purified. An equal amount of heat inactivated Trypsin or Protease K (inactivated at 95 °C for 1 h) were used as controls. The treated OMVs were used for Western blot assay and anti-*Plasmodium* activity testing.

## Proteomic analysis of Su_YN1 OMVs by LC-MS/MS

Purified Su_YN1 OMVs in PBS solution were used for LC-MS/MS analysis. In-solution digestion of 50 µg OMV was conducted using trypsin overnight at 37 °C[64]. Peptide identification was performed on a Q Exactive mass spectrometer coupled to Easy nLC (Thermo Fisher Scientific). A C18-reversed phase column (15 cm long, 75 µm inner diameter) packed in-house with RP-C18 5 µm resin was used with buffer A (0.1% Formic acid in HPLC-grade water) and a linear gradient of buffer B (0.1% Formic acid in 84% acetonitrile) at a flow rate of 250 nl/min, controlled by IntelliFlow technology, over a 60 min period. The MS data were analyzed using MaxQuant software version 1.3.0.5[65]. The MS data were searched against the UniProt_Serratia_267125_20181228 database. The mass spectrometry proteomics source data were deposited to the ProteomeXchange Consortium via the PRIDE[66] partner repository with the dataset identifier PXD042831.

## Nanoparticle tracking analysis (NTA) and nanoflow cytometry measurement (NFCM)

Nanoparticle tracking analysis of Su_YN1 filtered culture supernatant (Fig. 1d) was 100-fold diluted and analyzed using NanoSight NS300 instrument (Malvern Panalytical) at 25 °C. Particle movement was analyzed by nanoparticle tracking analysis software following the manufacturer's instructions (NanoSight NS300 User Manual,

MAN0541-02-EN, 2018) with the minimal expected particle size, minimum track length and blur setting all set to automatic. Each video was then analyzed to determine the respective mean and distribution of OMV size[67]. Nanoflow cytometry measurement[68] (NFCM) was used to analyze particle size and particle proportion of PBS-resolved purified OMVs from Su_YN1 cultured in serum from various sources (OMVs purified from 26 mL culture resolved in 1 mL PBS). NFCM was also used to conduct fluorescence nanoparticle analysis to estimate the proportion of NBD-PC incorporated into OMVs. The NFCM experiments were conducted by Neoland BioSciences Co., Ltd. Please notice that the size of OMVs may vary due to the different methods used.

## qRT-PCR analysis of *AmLip* transcription

Total RNA was extracted from $2 \times 10^9$ Su_YN1 bacterial cells at various time points in each group. Complementary DNA (cDNA) was synthesized from total RNA using the PrimeScript RT Reagent Kit with gDNA Eraser (Takara RR047B) according to the manufacturer's instructions. A quantitative real-time PCR (qRT-PCR) analysis was performed with the PikoReal 96 (Thermo) using the AceQ qPCR SYBR Green Master Mix (Vazyme Q111-03). The *Serratia* stably expressed gene *gyrB* (Gene ID: NPGAP_00015) was used as ref. [69]. The primers for *gyrB* reference are: Forward 5′-GTATATCGGCGATACCGATGACG-3′; Reverse 5′-ATGATGACCTCTGCGGCTG-3′. The primers for AmLip are: Forward 5′-GAGGCGAAGGCGACATAGTTGGA-3′; Reverse 5′-CTATGCCGACGGC-TATACGCT-3′.

## Anti-*Plasmodium* activity assays of OMVs against *P. berghei* ookinetes

To assess the inhibition of *P. berghei* ANKA ookinete formation, high gametocytemia blood collected from infected ICR mice (six-week-old male, approve number: A01MP2001) was used for in vitro ookinete culture[70]. Filtered culture supernatant or OMV solutions were added to ookinete culture. After culture at 20 °C for 20 h, ookinetes in the control and test groups were counted, and the inhibition rate was calculated refer to the ookinete formation rate of control as 100%.

## OMV loading with recombinant AmLip protein

*AmLip*-KO mutant Su_YN1 bacteria were cultured in RPMI 1640 medium with 10% FBS for 6 h and recombinant AmLip protein, or AmLip protein fragments were added at final concentration 10 µg/ml. Control was in RPMI 1640 medium with 10% FBS without bacteria, but containing equal amount of recombinant AmLip protein. After culture for another 20 h, culture supernatants were collected and OMVs were purified for Western blot assay and anti-*Plasmodium* activity test. Alternatively, 20 µg recombinant AmLip protein (6His tagged) was incubated with equal amount of 100 µg purified *E. coli* W3110 K12 OMVs and AmLip-KO Su_YN1 OMVs for 6 h and purified for Western blot assay.

## Protein extraction and Western blot analysis

Protein extraction from *Serratia* Su_YN1 bacteria, 293T cells, or MSQ43 cells was performed using RIPA buffer (R21237, Yuanye Biotech) supplemented with a complete protease inhibitor cocktail and 1 mM PMSF. After ultrasonication, the protein solution was centrifuged at 10,000 g for 15 min at 4 °C. The resulting supernatant was collected, and Laemmli sample buffer (PG112, Yamei Biotech) was added. For protein extraction from culture supernatant, Laemmli sample buffer was added directly to the filtered culture. For protein extraction from OMVs, Laemmli sample buffer was added to the purified OMV solution, followed by ultrasonication. The extracted protein samples were fractionated by 10% SDS-PAGE and transferred to a PVDF membrane. The membrane was then blocked with blocking buffer (5% BSA in TBST) and incubated with primary antibodies,

including AmLip mouse antiserum (as previously described[3]) at a dilution of 1:1000; rabbit anti-HA antibody (CST, cat#3724 S) at a dilution of 1:1000; mouse monoclonal Anti-His tag antibody (ab18184, Abcam) at a dilution of 1:1000; anti-CD63 rabbit monoclonal antibody (A22343, ABclonal) at a dilution of 1:500; anti-CD9 mouse monoclonal antibody (sc-59140, Santa Cruz) at a dilution of 1:500. After incubation with the primary antibodies, the membrane was washed three times with TBST and then incubated with HRP-conjugated secondary antibodies (ab6789, ab6721, Abcam). The membrane was washed four times in TBST prior to enhanced chemiluminescence detection (SQ201, Yamei Biotech).

### *P. berghei* standard membrane feeding assays (Pb SMFAs)

The procedure for *P. berghei* Standard Membrane Feeding Assays (Pb SMFAs) used in this study was adapted from the previously published method[25], and a workflow diagram illustrating the procedure is provided in Supplementary Fig. 8. To successfully conduct Pb SMFAs and achieve reliable infections in mosquitoes, it is crucial to avoid a significant drop in blood temperature drop during colleciton and carefully control the amount of anticoagulant heparin used. The heparin solution was prepared at a concentration of 10 mg/ml in PBS as a stock solution. To collect blood, 10 μl of the heparin solution was added to a 1.5 ml tube and thoroughly mixed by vortexing. The tube was then pre-warmed in a dry thermostat to 37 °C prior to blood collection. One tube was used to collect blood from each mouse, typically amounting to 1–1.3 ml of blood. The brief procedure for Pb SMFAs is as follows: 1) set up and pre-warm the membrane feeding devices, as well as the tubes containing the samples to be tested (control PBS and OMVs solution added in RPMI 1640) in a dry thermostat set to 37 °C; 2) pre-warm the tube containing heparin to collect blood in a dry thermostat at 37 °C, and rapidly collect mouse blood in the pre-warmed tube; 3) immediately transfer the collected blood to the tube containing the desired sample, mixed thoroughly, and promptly add it to the membrane feeding device connected to a circulator bath set at 38 °C for feeding; 4) mosquito blood-feeding was conducted in a room with environmental temperature set at 21 °C. Mosquitoes were allowed to feed blood for 15–20 min.

### Effect of OMVs on oocyst formation by membrane feeding assay

To test inhibitory activity of OMVs against *P. berghei* ANKA oocysts, we followed the "*P. berghei* Standard Membrane Feeding Assays (Pb SMFAs)" procedure[25]. Briefly, infectious blood collected from *P. berghei* ANKA-infected mice were collected and 2-fold diluted in 37 °C pre-warmed RPMI 1640 medium containing purified OMVs at final concentration 100 μg/ml, or 37 °C pre-warmed RPMI 1640 medium containing PBS (set as control) and immediately fed to mosquitoes through a membrane feeder. The control group (PBS added) and the test group (OMVs solution added) mosquitoes were all fed through membrane feeding device using blood collected from the same mice at the same time under the same conditions. The engorged mosquitoes were maintained at 22 °C and 75 ± 5% RH. Their midguts were dissected on day 8 after the blood meal, oocyst loads were counted using a fluorescence microscope.

### Effect of OMVs on ookinete motility

In vitro cultured *P. berghei* ANKA ookinetes were washed with RPMI 1640 medium, incubated with OMVs at final concentration of 100 μg/ml at 25 °C for 5 min, and then embedded in Matrigel (356234, BD Biocoat) for ookinete motility tracking as previously described[70]. More than 15 ookinetes were monitored for 15 min to determine the distance of migration, and to calculate the motility speed.

### Anti-*Plasmodium* effect of OMVs on asexual-stage *Plasmodium falciparum*

*P. falciparum* 3D7 asexual stage parasites were incubated with Su_YN1 OMVs at final concentration of 100 μg/ml at 37 °C for 5 min, washed

with RPMI 1640 medium to remove the OMVs, then adjusted to 0.1 % parasitemia and cultured for another 4 days. Parasite replication was monitored through Giemsa-stained blood smears and parasitemia was counted on day 2 and day 4 after culture.

### OMV staining and uptake by malaria parasites

DiO (DiOC18(3), C1038, Beyotime) or DiI (DiIC18(3), E607331-0100, Sangon Biotech) was added to the culture medium (at final concentration of 5μM) after Su_YN1 had been cultured for 12 h, and then cultured for another 12 h prior to culture supernatant filtration and OMVs purification. The labeled OMVs were co-cultured with *P. berghei* ANKA parasites (asexual stage and zygote culture) for 10 min and the uptake of OMV was observed using a fluorescence microscope.

### Living tracking and OMV uptake assay by *Plasmodium falciparum* asexual parasites

DiI-stained OMVs were added to *P. falcuparum* 3D7 asexual stage parasites at final concentration of 100 μg/ml, placed in a glassed bottom culture dish (YA0572, Solarbio) and monitored under living conditions using an inverted TCS SP8 confocal microscope (Leica Microsystems). Time-lapse images were taken at 30 sec intervals for 10 min. These images were also used to generate videos showing parasite lysis by OMVs.

### Immunofluorescence assay of OMVs and OMVs-treated parasites

Immunofluorescence assay of OMVs followed established protocols for vesicle-protein detection experiments[71,72]. The OMVs were first pre-stained with DiI, and then purified and washed to remove any unbound DiI dye. The stained OMVs were subsequently resuspended in PBS and incubated with a Poly-D-lysine treated cover slip to facilitate their attachment to the surface of the cover slip. After the attachment of OMVs on the cover slip, immunofluorescence staining was performed by immunolabeling the OMVs with AmLip antiserum and Alexa 488-conjugated goat anti-mouse IgG. For the immunofluorescence assay of OMVs-treated parasites, *Plasmodium* parasites of different stages treated with OMVs (unstained or pre-stained with DiO orDiI) were washed with PBS, and fixed using freshly prepared 4% paraformaldehyde (P6148, Sigma-Aldrich) in PBS for 15 min at room temperature and transferred to a 24-well cell plate containing a poly-l-lysine (E607015, Sangon Biotech)-pretreated coverslip at the bottom. The fixed cells were permeabilized with 0.1% Triton X-100 PBS solution for 5 min at room temperature, washed with PBS three times, blocked in 5% bovine serum albumin (BSA) solution for 60 min at room temperature, and incubated with the primary antibodies (AmLip mouse antiserum, EXP1 rabbit antiserum, or FITC-Phalloidin (YP0059S, UelandyInc) diluted in 5% BSA-PBS at 4 °C for 12 h. After wash with PBS three times, the coverslip was incubated with fluorescent-conjugated secondary antibodies (Alexa 488 conjugated goat anti-mouse IgG antibody, A11001; Alexa 555 conjugated goat anti-rabbit IgG antibody, A21428) for 1 h at room temperature and washed with PBS three times. Cells were stained with Hoechst 33342 (#62249, Thermo Fisher Scientific), mounted in 90% glycerol solution, and sealed with nail polish. All images were captured and processed using identical set-tings on a Leica SP8 confocal microscopy.

### OMV lipid extraction and lipidome analysis

Total lipids of OMVs purified from different origins (*E.coli* K12 cultured in LB, and Su_YN1 cultured in LB, Su_YN1 cultured in RPMI1640 with or without FBS) were extracted following standard procedures of lipid extraction by Bligh and Dyer method[73]. Briefly, OMV solutions in PBS were mixed with 1 volume of chloroform and 2 volumes of methanol. After thoroughly vortexing, 1 volume of chloroform was added followed by another mixing step. Afterwards, 1 volume of distilled water was added and vortex again. The tube was then

centrifuged at 2000 g for 20 min to facilitate complete phase separation, and the lower phase was collected for lipidome analysis using Q Exactive UHMR Hybrid Quadrupole-Orbitrap Mass Spectrometer. Lipids were separated in a Acclaim C30 column (with column temperature set at 50 °C, flow speed 0.26 ml/min) under following liquid chromatography conditions: mobile phase A (ACN:H2O, 60:40, v/v, containing 2 mM ammonium formate), mobile phase B (IPA:ACN, 90:10, v/v, containing 2 mM ammonium formate), with mobile phase B started at 30% and gradually increased to100%. The source lipidome data was processed using MS-DIAL software (ver.4.38). The processed data was further analyzed by the Clustvis (ClustVis: a web tool for visualizing clustering of multivariate data (BETA) (http://biit.cs.ut.ee/clustvis/)[74].

### Antagonization of phospholipids in OMV's anti-*Plasmodium* activity

Lipid antagonization test was modified from a previous reported method[32]. Briefly, Phosphatidylcholine (#L0023, TCI), phosphatidylethanolamine (GC44631-5, GLPBio), and Phosphatidylserine (S20166, Yuanye Biotech) dissolved in chloroform (500 μg /ml) were added to a glass-bottom 48-well culture plate (P24-1.5H-N, Cellvis). Phospholipids solutions were used to coat the culture dish to achieve 156 μg/cm$^2$, 312 μg/cm$^2$ and 625 μg/cm$^2$. Chloroform solvent was used as control. After evaporation of the chloroform solvent, 250 μl the RPMI 1640 *Plasmodium* culture medium containing 0.5% Album Max I (11020-021, Gibco) was added to the dish and incubated at 37 °C for 30 min. After incubation, *Pb NAKA-GFG/Luciferase* asexual parasites collected from ICR mice were diluted in 37 °C pre-warmed RPMI 1640 medium to 4% RCT and 250 μl of the parasite cultures were divided into the plate wells, and cultured for 1 h. OMVs were then added to the culture wells (final concentration of 100 μg/ml) and cultured for another 5 h. Parasite viability was determined by monitoring the bioluminescent signal intensity (RLUs) after the addition of 100 μg/ml substrate D-luciferin (MX4603-100MG, MKBio).

### Phospholipids antagonize OMV uptake analysis

*P. falciparum* 3D7 asexual parasites cultured in phospholipids bottom-coated culture plate were prepared as described above and collected. DiI stained OMVs were added to the parasites and incubated at 37 °C for 5 min. The parasites were then stained with a fluorescent DNA stain Hoechst 33342 (62249, Thermofisher) and immediately monitored under physiological living conditions using an inverted confocal microscopy Leica SP8.

### PC quantification assay and OMV incorporation test

The PC level was quantified using a phosphatidylcholine quantification kit (Sigma-Aldrich MAK049) following the provided instructions. For PC quantification, 100 μl of culture supernatant, 1 OD of bacteria, or 100 μg of OMV lysate in PC assay buffer were used. A sample blank without PC hydrolysis enzyme was included to subtract the background. The results were obtained using a fluorometric microplate reader, measuring the fluorescence at Ex/Em: 535/587 nm. To detect PC incorporation by Su_YN1 into OMVs, 1 mM Nitrobenzoxadiazole labeled Phosphatidylcholine (NBD-PC, Avanti Polar Lipids, #810132C) was added to the Su_YN1 culture containing 10% FBS and cultured for 12 h. OMVs were purified by ultracentrifugation. The incorporation of NBD-PC was visualized either under a stereo fluorescence microscopy or measured using a fluorometric microplate reader at Ex/Em: 467/539 nm.

### Flow cytometry analysis

*P. falciparum* 3D7 asexual parasites were incubated in RPMI 1640 medium (without Album Max I) containing 3 mM Nitrobenzoxadiazole labeled Phosphatidylcholine (NBD-PC, Avanti Polar Lipids, #810132 C) at 37 °C for 15 min. The NBD-PC treated (and not treated control)

parasites were then stained with Hoechst 33342 and incubated with DiI stained OMVs for 2 min, and data was collected using SONY flow cytometer SH800S. The parasites (Hoechst positive populations) were gated out and DiI signal was monitored and displayed as histogram. Data was analyzed using FlowJo (V10.8.1).

### *Plasmodium* PC scavenging inhibition analysis

To block *Plasmodium* PC scavenging, the parasites were either incubated at 4 °C for 15 min, treated with 1 mM NaN3 (S2002-5G, Merck) for 6 min at 37 °C to deplete ATP, or treated with 10mM L-α-GPC (G5291, Merck) on ice for 30 min before recovering to room temperature as previously described[37]. The treated parasites were then incubated with 3 mM NBD-PC or DiI-stained OMVs for 5 min and immediately monitored under living conditions using an inverted TCS SP8 confocal microscopy (Leica Microsystems).

### Statistical analysis

The statistical significance of the difference in oocyst number and ookinete motility speed was analyzed using a two-tailed Mann–Whitney test. The statistical significance of multiple groups comparisons was analyzed using one-way ANOVA test. The statistical significance of the effect of various modified OMVs on *Plasmodium* killing at different concentrations was analyzed using two-way ANOVA test. Other statistical significance was calculated using two-tailed Student's *t* test. A value of $P < 0.05$ was regarded as statistically significant difference. All statistics were performed using GraphPad Prism version 5.00 for Windows (GraphPad Software).

### Reporting summary

Further information on research design is available in the Nature Portfolio Reporting Summary linked to this article.

## Data availability

The data supporting the findings of this study are available in the article and the Supplementary Information. Source data are provided with this paper. The Su_YN1 OMVs mass spectrometry proteomics data generated in this study have been deposited to the ProteomeXchange Consortium via the PRIDE partner repository under accession code PXD042831. Source data are provided with this paper.

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

## Acknowledgements

This work was supported by grants from the National Natural Science Foundation of China (grants 31830086, 32000348 and 32021001), and NSFC/BMGF joint Grand Challenges programs (82261128007 and 2022YFML1006), the National Key R&D Program of China (grant 2019YFC1200800), Chinese Academy of Science (317GJHZ2022028GC) and Youth Innovation Promotion Association CAS. We are grateful to Marcelo Jacobs-Lorena at Johns Hopkins University School of Public Health for comments and proofreading the manuscript. We thank Fang Li for rearing mosquitoes. We thank Shen Yang for sharing with us *E. coli* K12 strain. We thank Xiaoyan Xu and Xingru Liao from the Core Facility Centre, CAS Centre for Excellence in Molecular Plant Sciences for assistance in lipidome analysis, Xiaoyan Gao, Jiqin Li, Zhiping Zhang and Lina Xu for technical support with electron microscopy, Wenjuan Cai for technical support with confocal microscopy. We are grateful to Malvern Panalytical company for assisting nanoparticle tracking analysis. We thank Haoran Wang and Junfeng Wang from Neoland BioSciences Co., Ltd for assisting nanoflow cytometry measurement (NFCM). We thank Qiyu Chu from Shanghai hoogen biotech co.ltd for assisting OMV proteomic analysis.

## Author contributions

S.W. and H.G. conceived the project. S.W., H.G. and Y.J. designed the study. Y.J. and H.G. conducted mosquito gut section and TEM analysis. Y.J. and H.G. conducted IEM analysis. H.G. and Y.J. conducted SEM analysis. L.W., H.G., Y.J. and W.H. conducted OMV isolation, purification, and quantification analysis. H.G. and Y.J. conducted OMV nanoparticle tracking analysis. H.G., L.W. and Y.J. conducted western blot experiments. H.G., Y.J. and L.W. conducted OMV protease digestion assay and recombinant AmLip protein binding test. H.G., Y.J. and L.W. conducted anti-*Plasmodium* activity test. H.G., Y.J. and G.W. conducted ookinete motility test, and examined the killing effect of OMVs on oocyst and sporozoite. H.G. L.W. and Y.J. conducted live-cell targeting and tracking of OMVs in parasites and indirect immunofluorescence assays. H.G. L.W. Y.J. and W.H. conduct OMV lipidome analysis. H.G., Y.J. and L.W. conducted the lipid antagonization assays. H.G. and Y.J. conducted flow cytometry analysis and PC scavenging pathway inhibition assays. D.L. provided technical help in living tracking experiments and indirect immunofluorescence assays. H.G. and S.W. analyzed the data. H.G. and S.W. wrote the manuscript.

## Competing interests

The authors declare no competing interests.
