## [Peer Review File · Nature Communications]

nature portfolio

Peer Review FileReviewer #1 (Remarks to the Author):

The authors previously discovered that a member of the mosquito gut microbiota, *Serratia ureilytica* Su_YN1, produces a lipase (AmLip) that inhibits the development of *Plasmodium* parasites in the mosquito. They aim to utilize this microbe to decrease incidences of malaria without harming mosquito populations. In this manuscript, the authors investigate the mechanism by which AmLip is transported to *Plasmodium* parasites in the context of the mosquito midgut. They propose that AmLip is secreted into the extracellular space where it associates with outer membrane vesicles (OMVs) produced *S. ureilytica* and is taken up by the parasites via the phosphatidylcholine scavenging pathway where AmLip becomes active and kills the parasites. The results are very intriguing, however I have several concerns about this work

1) According to the model the lipase is secreted by a T1SS to then interact with OMV. Is this interaction required? The authors never tested purified AmLip for anti-*Plasmodium* activity in the provided experiments. In addition, their model is that AmLip is "preferentially loaded onto OMVs" after secretion into the extracellular space. However, experiments to show how AmLip is preferentially loaded onto OMV have not been presented. Due to their composition, it is not uncommon for OMVs to nonspecifically associate with extracellular biomolecules. A demonstration that a specific region of the lipase specifically interacts with OMV would strengthen this manuscript.

2) The figures presented in figure 1 are unclear. How do the authors know that the structures visualized are of bacterial origin? There are several structures of irregular shape, that could indicate lysis. How do the authors define what an OMV is? How do they differentiate between OMV and eOMV? Gram-negative bacteria are damaged, and some will lyse as part of the normal process. These lysis byproducts will co-precipitate with OMVs produced by live cells. This could especially be a concern in the current study because they are taking samples from the mosquito midgut for their in vivo experiments and adding FBS to cultures for in vitro studies, and both methods could result in significant contamination with eukaryotic vesicles. In fact, I am not convinced that the material seen in figure ext 1 corresponds to OMV.

In addition, since the authors are conducting experiments in complex environments where there could be contamination with eukaryotic vesicles, it could be useful for the authors to blot for exosomal markers, like CD63, CD9, etc. This way they can verify the origin of the vesicles in their preparations to ensure that they are derived from their bacterium.

3) A proteomic analysis of OMV would strengthen this work. It would be nice to see only the lipase attached to OMV and not other unspecific T1SS effector attached to them.

4) Loading for western blots is inconsistent. Figure 2A needs a loading control for the total bacteria panel because AmLip appears to be expressed significantly less in the FBS- condition, which could explain why it is not detected in the supernatant. Figure 2B does not have consistent loading for each lane. The authors stated that the two supernatant lanes contain 20uL of sample, while the OMV lane contains 20ug of protein. The authors must be very clear and consistent regarding how they standardize their samples because it is not possible to make a comparison the way the experiment has been presented.

5) In figure 4C, the authors only treat infected red blood cells with OMVs, but never treated uninfected cells with OMVs.

6) Extended data figure 2B should compare growth curves of *S. ureilytica* in the presence and absence of FBS to prove there is no impact on fitness.

7) Extended data figure 8B is difficult to interpret because the authors only supplement with FBS for one condition. Most bacteria do not make phosphatidylcholine, but this lipid is abundant in eukaryotes. Because of this, the authors are likely just quantifying phosphatidylcholine present in the FBS.

Minor

Notation for truncated AmLip is not correct. Authors made a truncated version of AmLip that lacks the T1SS secretion signal and refer to it in the manuscript as AmLip Δ T1SS. This nomenclature is confusing because these strains are not lacking the T1SS itself, they are just truncated versions of AmLip. The current experiment provides information, however, the truncated AmLip could have an

altered protein structure which prevents it from being secreted that is unrelated to the T1SS. A clearer experiment would be to make a deletion in the T1SS to show that secretion of the effector is T1SS-dependent.

Reviewer #2 (Remarks to the Author):

The manuscript titled "Outer membrane vesicles from a mosquito commensal mediate targeted killing of Plasmodium parasites via the phosphatidylcholine scavenging pathway" presents interesting findings of a bacterial effector that binds to bacterial OMVs to be translocated and interact with malaria parasites. They show that OMV AmLip affect the membrane integrity of ookinetes and malaria parasites, providing further proof that extracellular vesicles work in cross-kingdom interactions. The manuscript is well written and some of the findings are very exciting. Nevertheless, some of the experiments and statistical analysis need to be refined to demonstrate more conclusively some of the findings presented in the manuscript. Some of the concerns of the reviewer are presented below:

Major concerns

- 1- The authors are presenting contradicting results that invalidate the model presented in Figure 2i. According to the model developed by the authors and the results from Figure 2 a host serum induces the production of AmLip (protein). However, in extended figure 3a they clearly showed that transcription is not affected by the addition of serum. If there are no transcripts, how do the authors explain that there is more protein? Do the authors have a protein that can serve as a loading control to show that this is not due to problems with loading or transfer?
- 2- Several of the experiments should be analyzed as ANOVA (either one or two-way). However, the authors do all the comparisons through t-tests. This is cherry picky as they are just analyzing certain pairs and takes part of the variability from the experiment away. Whereas in some experiments pairwise analysis may be ok to perform. Other experiments need to be properly analyzed. For example, in figure 2 experiment presented in 2e should be an ANOVA. G should be analyzed through a two-way ANOVA and h is one-way ANOVA. Post-hoc analysis should also be performed to define differences between groups. Please consult a statistician to decide the appropriate statistical test.
- 3- What is the biological relevance of the estimation developed by the authors in Ext. Data Fig 5? It is likely that there are significant differences between the secretion of vesicles in vivo within the midgut of a mosquito versus in a culture. The authors are mixing two systems (in vivo and in vitro) to come up with an equation that most likely do not accurately (or vaguely) estimates the number of vesicles secreted by the bacteria in the midgut. If the point was to estimate the number of vesicles to validate the downstream experiments, the authors should have used the images from their TEM experiments to estimate the number of vesicles in different areas through the midgut of the mosquito (and using multiple mosquitos). This will give a most accurate estimation of the number of vesicles. Given that they have access to an NTA, they could have used OMV preparations with a biologically relevant number of vesicles.
- 4- In figure 4d, several of the OMVs do not show double labeling. AmLip (green) particles are observed without DiL signal (red) [no yellow]. What are the green only particles? It is expected that not all the OMVs will contain the AmLip, but according to your results, there is very little free AmLip. Please provide a quantification of the actual colocalization of AmLip and the DiL in the vesicles. This can be done with the OMVs only without incubating with the parasites. This will also provide an idea of the heterogeneity of the OMV population.
- 5- The mechanism of PC incorporation into the OMVs is very unclear and poorly demonstrated. The authors claim that "very small amounts of PC were detected in Su-YM1 cells". Yet, they show that labeled PC from the culture is incorporated into the OMVs. Which brings up the question, how biologically relevant is Figure 5c? Were the OMVs purified from one OD of bacteria? Otherwise, they are over-representing the levels of PC in the OMVs and under-representing in the bacteria. What is the level of NBD-PC incorporated into the bacteria that produced the vesicles in Ext. Dat. Fig. 8? The authors can show a picture of the bacteria with the lipids. Also, does incubation of the bacteria with PC alone induce the production of OMVs to similar levels that serum does? Further, are all OMVs positive for NBD-PC? It is likely that the incorporation of PC into the membrane of the

bacteria leads to the invagination of the membrane in a process like the biogenesis of microvesicles in Eukaryotic cells. The authors could investigate if the bacterial membrane is deformed with the incorporation of these lipids.

6- In toxoplasma, lipid scavenging requires endocytosis likely mediated by receptors. By coating the plates with different lipids, the authors immobilized them in the plate, which would limit the interaction with receptors. Competition assays should be performed by incubating the parasites with free lipids as done in Ext. Data. Fig. 9d.

Minor concerns

1- Figure Ext Dat Fig 6a and Ext Dat Fig 8 do not show how efficiently the OMVs are labeled. The authors can either use the NTA to do this (if there Nanosight has the filters for fluorescence detection). Labeled versus unlabeled vesicles can be quantified and differences can be measure. Otherwise, just remove the figures. They are not informative and does not show anything.

2- Why is there so much DiL signal in the control? Please include the percentage of DiL positive parasites and statistical analysis of the measurements.

Reviewer #3 (Remarks to the Author):

This study shows further analysis of *Serratia ureilytica* Su_YN1, a bacterium previously identified by the authors as secreting the anti-Plasmodium lipase protein AmLip. AmLip is incorporated into the cell membrane of Plasmodium and kills Plasmodium by hydrolyzing lipids. They revealed how effectors such as AmLip are selectively transferred to Plasmodium via OMVs. The stud is well-organized and provides new insights into the insect-bacteria-Plasmodium tripartite interaction.

The paper is worthy of publication in Nature Communications, but I would like to request a few points of clarification, as listed below.

(line 85)

Although the membrane vesicle-like structures in the midgut of blood-fed mosquitoes contain AmLip, there seems to be no direct evidence that these vesicles are of bacterial origin, in particular, that they are produced by Su_YN1. In vitro experiments using pre-cleared FBS confirmed that Su_YN1 can produce OMVs. It is recommended to prepare somehow vesicle-free blood and perform artificial blood-feeding experiments. (I understand that experiments using serum are difficult because mosquitoes hesitate to suck serum)

(line 170)

This is a comment related to the previous report by the authors published in Nature Microbiology. Is there any difference in the ability of AmLip alone (recombinant protein) and OMVs in which AmLip is incorporated to inhibit Plasmodium ookinete formation?

(line 185)

In each of the experiments in Figure 3, what would happen if AmLip protein alone was used without OMVs? Based on their findings, the presence of OMVs should promote the efficiency of AmLip in killing Plasmodium.

(line 223)

What is the possible mechanism by which OMVs can invade and destroy infected erythrocytes while not affecting normal erythrocytes? Please provide the readers with possible hypotheses.

(line 389)

What do you think is the advantage of midgut bacterium species such as Su_YN1 having the mechanism of OMVs production found by the authors?

Manuscript: NCOMMS-22-45996-T

Title: Outer membrane vesicles from a mosquito commensal mediate targeted killing of *Plasmodium* parasites via the phosphatidylcholine scavenging pathway

We appreciate all reviewers for the evaluation of our manuscript and for the constructive comments and suggestions. We have carefully revised the manuscript taking into account of all the reviewers' comments. Our point-by-point responses are provided below. Please note that reviewers' comments are quoted in bold and our responses follow in plain text.

Reviewer #1 (Remarks to the Author):

The authors previously discovered that a member of the mosquito gut microbiota, *Serratia ureilytica* Su_YN1, produces a lipase (AmLip) that inhibits the development of *Plasmodium* parasites in the mosquito. They aim to utilize this microbe to decrease incidences of malaria without harming mosquito populations. In this manuscript, the authors investigate the mechanism by which AmLip is transported to *Plasmodium* parasites in the context of the mosquito midgut. They propose that AmLip is secreted into the extracellular space where it associates with outer membrane vesicles (OMVs) produced *S. ureilytica* and is taken up by the parasites via the phosphatidylcholine scavenging pathway where AmLip becomes active and kills the parasites. The results are very intriguing, however I have several concerns about this work

Response: We greatly appreciate the Reviewer for the favorable comments.

1) According to the model the lipase is secreted by a T1SS to then interact with OMV. Is this interaction required? The authors never tested purified AmLip for anti-*Plasmodium* activity in the provided experiments. In addition, their model is that AmLip is “preferentially loaded onto OMVs” after secretion into the extracellular space. However, experiments to show how AmLip is preferentially loaded onto OMV have not been presented. Due to their composition, it is not uncommon for OMVs to nonspecifically associate with extracellular biomolecules. A demonstration that a specific region of the lipase specifically interacts with OMV would strengthen this manuscript.

Response: We appreciate the insightful comments. Our data suggest that the interaction of AmLip with OMV is required for robust anti-*Plasmodium* activity. Binding to OMVs strongly enhances the malaria targeting of AmLip, promoting its anti-*Plasmodium* activity. Notably, OMVs with AmLip kills parasites within several minutes (Figure 4B), while recombinant AmLip usually requires incubation for 6 hours, as we previously reported (*Nature microbiology*, 6(6), 806-817). Most secreted AmLip associates with OMVs, and the interaction of AmLip with

OMVs is not nonspecific. The interaction of AmLip with OMVs is strong and attributed to the special nature of this lipase family (E.C.3.1.1.3), which has high affinity and function at an oil-water interface, a phenomenon called interfacial activation (*Prog Biophys Mol Biol.* 2018 Jan;132:23-34). OMVs surface, especially these serum-induced OMVs containing substantial host-derived lipids, may provide such interfacial surface that favors lipase binding.

To demonstrate that specific regions of the lipase interact with OMVs, we expressed and purified full-length AmLip and AmLip fragments. The fragment 1 lacks the C-terminal ($\Delta 351-614$) that is required for recognition by T1SS, fragment 2 lacks the N-terminal lid structure ($\Delta 1-74$), while fragment 3 lacks both the C-terminal and N-terminal structures ($\Delta 1-74, \Delta 351-614$) (Figure A and B below). Only the full-length AmLip showed lipase activity on egg yolk plate (Figure C below). We incubated these fragments with Su_YN1 OMVs, purified and washed these OMVs, and then detected the binding of these AmLip fragments using anti-6His antibody. The result below indicate that only full-length AmLip efficiently binds OMVs (Figure D below), indicating that an intact protein structure is required for both lipase activity and OMV binding. We have included these data in Supplementary Fig. 3a-d of the revised manuscript.

Assay for binding of AmLip fragments to OMVs. (A) Schematic diagram showing AmLip protein structure and AmLip fragments. (B) Coomassie blue stained gel showing recombinantly expressed full-length AmLip protein and its fragments. (C) Lipase activity test of purified AmLip protein and its fragments on egg yolk plate. (D). Western blot analysis of OMV-bound AmLip protein fragments using anti-His antibody.

2) The figures presented in figure 1 are unclear. How do the authors know that the structures visualized are of bacterial origin? There are several structures of irregular shape, that could indicate lysis. How do the authors define what an OMV is? How do they differentiate between OMV and eOMV? Gram-negative bacteria are damaged, and some will lyse as part of the normal process. These lysis byproducts will co-precipitate with OMVs produced by live cells. This could especially be a concern in the current study because they are taking samples from the mosquito midgut for their *in vivo* experiments and adding FBS to cultures for *in vitro* studies, and both methods could result in significant contamination with eukaryotic vesicles. In fact, I am not convinced that the material seen in figure ext 1 corresponds to OMV.

Response: In addition to the IEM results shown in Figure 1, we employed several additional methods to confirm the biogenesis of OMVs by Su_YN1 both *in vivo* in the mosquito gut and *in vitro* culture. We examined membrane vesicles produced by Su_YN1 in the gut using TEM (Fig. 1b, upper panel) and SEM (Fig. 1b, lower panel). We ripped open mosquito guts containing only Su_YN1 bacteria and detected OMV biogenesis using cryo-EM (Figure A below). We confirmed that these OMVs are mainly produced by living Su_YN1 bacteria but not eOMVs released during bacteria death. Firstly, we can clearly observe budding of OMVs from bacteria outer membrane in these electron microscope images. Secondly, Su_YN1 bacteria grow well and remain highly viable both *in vivo* and *in vitro* (Extended Data Fig2, and following response 6#). To avoid contamination by eukaryotic vesicles, FBS was pre-cleared by ultracentrifugation to remove the eukaryotic vesicles before use in the *in vitro* studies (Extended Data Fig1). Notably, Su_YN1 produces massive OMVs upon induction, and the extremely high yield of Su_YN1 OMVs is easily distinguished from the background of bloodmeal lysis products.

To further address the reviewer's concern and as suggested by reviewer 3: "*it is recommended to prepare somehow vesicle-free blood and perform artificial blood-feeding experiments*", we fed Su_YN1-carrying mosquito with EV-free FBS and observed OMVs in the mosquito gut using TEM. We observed OMVs only in the guts of Su_YN1-carrying mosquito fed with FBS, but not in axenic mosquito fed with FBS or Su_YN1-carrying mosquito fed with a sugar meal (Figure B below). The figures in Extended Data1 show OMVs in the culture supernatant that were directly stained using phosphotungstic acid. Here we provided nanoflow cytometry measurement (NFCM) (by Neoland BioSciences Co.,Ltd) of the OMVs purified from the culture supernatant of Su_YN1 cultured with various serum sources (Table below) The size of particle varies due to the detection method. High proportions of OMVs are detected in the supernatant of Su_YN1 cultures using various serum resources. We have added these new results to the revised manuscript.

Detection of Su_YN1 OMVs in the mosquito gut. (A) Cryo-scanning electron microscopy image of the midguts of *An. stephensi* carrying Su_YN1 36 h after a blood meal. Blue arrow heads indicate Su_YN1 bacteria. Red arrow heads indicate outer membrane vesicles (OMVs). The scale bar represents 200 nm. (B) Transmission electron microscope (TEM) images of Su_YN1 OMVs from the mosquito midgut lumen. *Anopheles* mosquitoes carrying Su_YN1, or axenic *Anopheles* mosquitoes, were fed with either FBS or a sugar meal, and 36 hours later, their midguts were sectioned for TEM analysis. The white arrow heads indicate OMVs. Scale bar, 1 μ m.

Table. Particle analysis of OMVs from Su_YN1 culture in the presence of serum from various sources

Sample	particle size (nm)	Particle concentration (/ml)	Vesicle portion
Human serum mock purified	Unable to detect.		
Human serum induced OMVs	82.50 ±15.85	2.78E+11	99.05%
Bovine serum mock purified	Unable to detect.		
Bovine serum induced OMVs	82.37 ±14.47	3.29E+11	98.44%
Mouse serum mock purified	Unable to detect.		
Mouse serum induced OMVs	80.06 ±14.93	1.86E+11	86.78%
Porcine serum mock purified	80.76 ±16.70	1.21E+08	86.49%
Porcine serum induced OMVs	82.00 ±15.29	1.66E+11	89.28%

In addition, since the authors are conducting experiments in complex environments where there could be contamination with eukaryotic vesicles, it could be useful for the authors to blot for exosomal markers, like CD63, CD9, etc. This way they can verify the origin of

the vesicles in their preparations to ensure that they are derived from their bacterium.

Response: We performed IEM on Su_YN1 OMVs from the mosquito gut section using CD63 antibody (Cat# A22343, ABelonal) and AmLip antiserum. We did not detect any positive signals of CD63 on Su_YN1 OMVs, whereas AmLip was detected (Figure below). We have included this data in Supplementary Fig. 1c in the revised manuscript.

IEM of Su_YN1 OMVs in the mosquito gut. We used immune electron microscopy (IEM) to detect CD63 and the lipase AmLip on the OMVs of Su_YN1 in the midgut of *Anopheles* mosquitoes. An ultrathin cryosection of the midgut was analyzed, and positive staining was indicated by white arrowheads. The scale bar is 100 nm.

3) A proteomic analysis of OMV would strengthen this work. It would be nice to see only the lipase attached to OMV and not other unspecific T1SS effector attached to them.

Response: Thank you for your suggestion to conduct a proteomic analysis of OMV to further strengthen our work. We have taken your advice and performed a proteomic analysis of purified Su_YN1 OMVs. The most abundant proteins (ranked by LFQ intensity) results are presented in a table below, which shows that shows the lipase AmLip is highly abundant in the OMVs (highlighted in grey). However, T1SS effectors are not found in the entire list of OMVs proteomic analysis. We have added this data to Supplementary Fig.2 in the revised manuscript.

Table. Proteomic analysis of Su_YN1 OMVs.

Protein names	Peptides	LFQ intensity
Preprotein translocase subunit YajC	3	20883000000
Serine 3-dehydrogenase	23	17159000000
Uncharacterized protein	1	15225000000
Autotransporter outer membrane beta-barrel domain-containing protein	25	13735000000
Flagellin	24	11796000000
Serralysin	18	11724000000
Lipase (AmLip)	18	10028000000
Glutathione ABC transporter ATP-binding	1	6334100000

protein		
Major outer membrane lipoprotein	5	4258600000
N-acetylmuramoyl-L-alanine amidase	10	4225900000
Outer membrane protein F	14	3520800000
Phage tail protein	19	3380800000
Polyurethanase	12	2546400000
Ligand-gated channel protein	28	2029000000
Phage major tail tube protein	5	2007400000
Outer membrane protein C	8	1758000000
Type VI secretion system tip protein VgrG	14	1748400000
Hemolysin	60	1492800000
Putative acetyltransferase	1	1453200000
Porin OmpA	12	1281500000
TonB-dependent receptor	20	1007900000
Uncharacterized protein	9	1001900000
Uncharacterized protein	1	782060000
TonB-dependent heme/hemoglobin receptor family protein	4	690080000
Terminase	1	661100000
Uncharacterized protein	5	649570000
Patatin	10	631840000
Phage tail protein	12	616930000
GntR family transcriptional regulator	1	614300000
Acetyltransferase component of pyruvate dehydrogenase	7	601090000

4) Loading for western blots is inconsistent. Figure 2A needs a loading control for the total bacteria panel because AmLip appears to be expressed significantly less in the FBS- condition, which could explain why it is not detected in the supernatant. Figure 2B does not have consistent loading for each lane. The authors stated that the two supernatant lanes contain 20uL of sample, while the OMV lane contains 20ug of protein. The authors must be very clear and consistent regarding how they standardize these samples because it is not possible to make a comparison the way the experiment has been presented.

Response: In Figure 2A, the two lanes (FBS- and FBS+) of the bacteria samples show total proteins that were equally extracted from 1 OD of bacteria. We have now included the loading control, which is a commissive blue staining of total bacteria protein (Figure A below). This data has been added to Extended Data Fig. 3b in the revised manuscript. In figure 2B, we standardized the experiment by resolving OMVs in PBS at a volume comparable to the initial volume of the supernatant, and we took an equal volume for western blot analysis. The new result shows enrichment of AmLip in OMVs fraction (Figure B below). We have updated the

western blot in Figure 2B in the revised manuscript.

Detection of AmLip by western blot. (A) Commisive blue-stained SDS-PAGE gel of total Su_YN1 bacterial protein extracted from 1 OD bacteria. (B) Western blot detection of AmLip protein in Su_YN1 culture supernatant, OMV-depleted supernatant, and OMVs purified from the culture supernatant using AmLip antiserum. The OMVs were resolved in PBS at a volume comparable to the initial volume of the supernatant. Equal amounts of samples were taken for analysis.

5) In figure 4C, the authors only treat infected red blood cells with OMVs, but never treated uninfected cells with OMVs.

Response: Please note that the control (ctrl) lane in Figure 4C shows an image of uninfected red blood cells treated with OMVs for 30 minutes, and uninfected red blood cells do not bind OMVs.

6) Extended data figure 2B should compare growth curves of *S. ureilytica* in the presence and absence of FBS to prove there is no impact on fitness.

Response: In order to demonstrate the absence of an impact on fitness, we have included a comparison of the growth curves of *S. ureilytica* in the presence and absence of FBS. As shown in the figure below, Su_YN1 reaches a plateau after 20 hours in the absence of FBS (Figure below). However, in the presence of FBS, Su_YN1 shows even better growth. These results indicate that FBS is highly favored by this bacterium and does not have any impact on its fitness. The manuscript has been updated with the revised Extended Data Figure 2B.

Growth curve of Su_YN1 cultured with or without FBS. Su_YN1 bacteria were inoculated at a ratio of 1:100 in RPMI 1640 medium without FBS (FBS-) or RPMI 1640 medium supplemented with 10% (v/v) FBS (FBS+), and bacterial growth was monitored by measuring OD₆₀₀ over time. The right panel shows a microscopic image of Su_YN1 bacteria with high viability. Scale bar, 10 μ m.

7) Extended data figure 8B is difficult to interpret because the authors only supplement with FBS for one condition. Most bacteria do not make phosphatidylcholine, but this lipid is abundant in eukaryotes. Because of this, the authors are likely just quantifying phosphatidylcholine present in the FBS.

Response: Thank you for your comments. To clarify, the PBS control lane in original Figure 8b (see Extended data figure 8c in the revised manuscript) was prepared by mock purifying and washing RPMI 1640 plus 10% EV-free FBS+ medium using the same OMV purification procedure by ultracentrifugation that was used for the OMVs. The resulting “pellet” contained no material and was resolved in the same volume of PBS as the OMVs, to rule out any interference of the serum-containing medium condition. The OMVs were purified and washed with PBS to remove any remaining phosphatidylcholine and other factors from the medium could interfere with the results. We have revised the figure legend to provide a more detailed explanation of how the PBS control was set up.

Minor

Notation for truncated AmLip is not correct. Authors made a truncated version of AmLip that lacks the T1SS secretion signal and refer to it in the manuscript as AmLip Δ T1SS. This nomenclature is confusing because these strains are not lacking the T1SS itself, they are just truncated versions of AmLip. The current experiment provides information, however, the truncated AmLip could have an altered protein structure which prevents it from being secreted that is unrelated to the T1SS. A clearer experiment would be to make a deletion in the T1SS to show that secretion of the effector is T1SS-dependent.

Response: We thank the reviewer for pointing out. We have taken into consideration the reviewer's suggestion and have now revised the notation "AmLip Δ T1SS" to "AmLip Δ 351-614".

Reviewer #2 (Remarks to the Author):

The manuscript titled “Outer membrane vesicles from a mosquito commensal mediate targeted killing of *Plasmodium* parasites via the phosphatidylcholine scavenging pathway” presents interesting findings of a bacterial effectors that binds to bacterial OMVs to be translocated and interact with malaria parasites. They show that OMV AmLip affect the membrane integrity of ookinetes and malaria parasites, providing further proof that extracellular vesicles work in cross-kingdom interactions. The manuscript is well written and some of the findings are very exciting.

Response: We greatly appreciate the Reviewer for the positive comments.

Nevertheless, some of the experiments and statistical analysis need to be refined to demonstrate more conclusively some of the findings presented in the manuscript. Some of the concerns of the reviewer are presented below:

Major concerns

1- The authors are presenting contradicting results that invalidate the model presented in Figure 2i. According to the model developed by the authors and the results from Figure 2 a host serum induces the production of AmLip (protein). However, in extended figure 3a they clearly showed that transcription is not affected by the addition of serum. If there are no transcripts, how do the authors explain that there is more protein? Do the authors have a protein that can serve as a loading control to show that this is not due to problems with loading or transfer?

Response: Thank you for bringing up this important point. The transcription of *AmLip* does not appear to be affected by the addition of serum. However, we would like to emphasize that our preliminary findings suggest that AmLip protein expression is post-transcriptionally regulated by a small RNA, which will be future investigated in our future study.

2- Several of the experiments should be analyzed as ANOVA (either one or two-way). However, the authors do all the comparisons through t-tests. This is cherry picky as they are just analyzing certain pairs and takes part of the variability from the experiment away. Whereas in some experiments pairwise analysis may be ok to perform. Other experiments need to be properly analyzed. For example, in figure 2 experiment presented in 2e should be an ANOVA. G should be analyzed through a two-way ANOVA and h is one-way

ANOVA. Post-hoc analysis should also be performed to define differences between groups. Please consult a statistician to decide the appropriate statistical test.

Response: We appreciate the reviewer's suggestions and have carefully checked and revised the statistical analysis as the reviewer suggested. We have changed the statistical tests used in Fig.1f, Fig.2e, Fig.2h, Fig.5c, Fig.5d, Ext.Dat.1c, Ext.Dat.3c, Ext.Dat.8c, Ext.Dat.9b, and Ext.Dat.9c to one-way ANOVA, while in Fig. 2g, we have used two-way ANOVA. Please see the revised manuscript for the updated statistical analysis.

3- What is the biological relevance of the estimation developed by the authors in Ext. Data Fig 5? It is likely that there are significant differences between the secretion of vesicles *in vivo* within the midgut of a mosquito versus in a culture. The authors are mixing two systems (*in vivo* and *in vitro*) to come up with an equation that most likely do not accurately (or vaguely) estimates the number of vesicles secreted by the bacteria in the midgut. If the point was to estimate the number of vesicles to validate the downstream experiments, the authors should have used the images from their TEM experiments to estimate the number of vesicles in different areas through the midgut of the mosquito (and using multiple mosquitos). This will give a most accurate estimation of the number of vesicles. Given that they are access to an NTA, they could have them use OMV preparations with a biological relevant number of vesicles.

Response: Thank you for your insightful comments. We acknowledge that there are significant differences between the secretion of vesicles *in vivo* within the midgut of a mosquito and *in vitro* culture. We agree with the reviewer that it is not accurate to estimate the number of vesicles in the midgut using the equation presented in Extended Data Figure 5. We did not estimate the number of vesicles using the equation in the downstream experiments. In light of these considerations, we have decided to remove the estimation presented in Extended Data Figure 5 to avoid any potential misunderstanding. Thank you for bringing this issue to our attention.

4- In figure 4d, several of the OMVs do not show double labeling. AmLip (green) particles are observed without DiL signal (red) [no yellow]. What are the green only particles? It is expected that not all the OMVs will contain the AmLip, but according to your results, there is very little free AmLip. Please provide a quantification of the actual colocalization of AmLip and the DiL in the vesicles. This can be done with the OMVs only without incubating with the parasites. This will also provide an idea of the heterogeneity of the OMV population.

Response: Thank you for your comment. We have now provided a separate image of the green and red channels, as well as a merged image showing colocalization of AmLip and DiL signals in the vesicles. We have also counted the dots with single or double labeling, as suggested (Figure A below). Our results show that the green or red labeling of OMVs colocalized well (Figure A below, right panel), but due to the limited focal plane and resolution of the confocal microscopy, and the discrepancy in fluorescence strength between the two labels, some puncta appeared either more green (see dot 1#) or more red (see dot 2#) (Figure A and B below).

Colocalization of AmLip with OMVs. Confocal images showing colocalization of AmLip (green channel) and OMVs (DiL stained, red channel). The signals are quantified in the right panel.

5- The mechanism of PC incorporation into the OMVs is very unclear and poorly demonstrated. The authors claim that “very small amounts of PC were detected in Su_YN1 cells”. Yet, they are show that labeled PC from the culture is incorporated into the OMVs. Which brings up the question, how biological relevant is Figure 5c? were the OMVs purified from one OD of bacteria? Otherwise, they are over-representing the levels of PC in the OMVs and under-representing in the bacteria. What is the level of NBD-PC incorporated into the bacteria that produced the vesicles in Ext. Dat. Fig. 8? The authors can show a picture of the bacteria with the lipids.

Response: The mechanism of PC incorporation into the OMVs is unknown, and we plan to explore this in our future study. We agree with the reviewer that direct comparison of the content of PC extracted from bacteria and the medium may cause confusion. To avoid confusion, we separately compare the PC content in Su_YN1 bacteria cultured in FBS- medium and FBS+ medium to show content of PC in the bacterial cells (Figure A below, Extended Data Fig.8b in the revised manuscript) We also separately compared the PC content of various media and resolved OMV solution to show the distribution of PC (Figure B below, Fig.5c in the revised

manuscript).

PC quantification and distribution assay. (A) Phosphatidylcholine quantification of Su_YN1 bacteria cultured in RPMI 1640 medium (FBS-) or RPMI 1640 plus 10% FBS medium (FBS+). Equal amounts of bacteria (1 OD) were used for PC extraction and detection. (B) Quantification of phosphatidylcholine (PC) in equal amounts (100 μ l) of RPMI 1640 plus 10% FBS medium (FBS+ medium), Su_YN1 supernatant cultured in FBS+ medium (Total supernatant), OMVs-depleted total supernatant (OMVs depleted), and OMVs fraction dissolved in comparable amount of RPMI 1640 medium (OMVs dissolved).

To directly show PC distribution in bacterial cells and the produced OMVs, we performed NBD-PC incorporation assay. We separated and compared the total Su_YN1 culture supernatant, the OMV-depleted supernatant fraction, bacteria fraction, and OMVs fraction from NBD-PC cultured Su_YN1. We found that the level of PC incorporated into the bacteria is significantly lower than the OMVs fraction (Figure A below). We also provide the fluorescence microscope images of Su_YN1 bacteria treated with or without NBD-PC, which show weak incorporation of PC into bacterium (Figure B below). Together, the majority of PC is incorporated into OMVs compared to bacterial cells.

NBD-PC uptake by *Serratia* Su_YN1 bacteria. (A) Fluorescence microscopy

images of various NBD-PC culture samples. Sample 1, Su_YN1 total culture supernatant; sample 2, OMVs-depleted Su_YN1 total culture supernatant; sample 3, Su_YN1 bacteria washed with PBS; sample 4, Purified OMVs solution in PBS. (B) Fluorescence microscopy images of Su_YN1 bacteria cultured with 3mM NBD-PC or without NBD-PC (Ctrl).

Also, does incubation of the bacteria with PC alone induces the production of OMVs to similar levels that serum does? Further, are all OMVs positive for NBD-PC? It is likely that the incorporation of PC into the membrane of the bacteria leads to the invagination of the membrane in a process like the biogenesis of microvesicles in Eukaryotic cells. The authors could investigate if the bacterial membrane is deformed with the incorporation of these lipids.

Response: We performed fluorescence nanoparticle analysis on NBD-PC--positives OMVs and found the majority of OMVs exhibited NBD signal (Figure A below), indicating efficient incorporation of PC into OMVs. However, PC alone does not strongly induce OMV biogenesis as observed with serum (Figure B below).

OMV incorporation and OMV induction assay of PC. (A) Fluorescence nanoparticle analysis of OMVs purified from Su_YN1 cultured without NBD-PC (Control), and OMVs purified from Su_YN1 cultured in the presence of 3mM NBD-PC. (B) Quantification of OMVs purified from Su_YN1 cultured in RPMI 1640 medium, RPMI 1640 medium supplemented with 10% FBS, or RPMI 1640 medium supplemented with 20 μ g/ml PC (corresponding to the physiological level of PC present in serum).

6- In toxoplasma, lipid scavenging requires endocytosis likely mediated by receptors. By coating the plates with different lipids, the authors immobilized them in the plate, which would limit the interaction with receptors. Competition assays should be performed by incubating the parasites with free lipids as done in Ext. Data. Fig. 9d.

Response: The competition assays presented in Figure 5D and Extended Data Fig 9C were adopted from similar experiments conducted by Nicolas M.B. Brancucci *et al* (*Cell*. 2017171(7):1532-1544). Lipids were dissolved in chloroform and added to a glass bottom plate. The solvent was evaporated before adding the parasite culture to prevent any harmful effect of organic solvents on the parasites during long-time culture, which is not a problem for experiment in Extended Data Figure 9d that does not require long-time incubation. Moreover, unlike *Toxoplasma* parasites that are relatively immobile during culture, *Plasmodium*-infected RBCs are unattached to the bottom of the plate and rotate constantly, which is driven either by the actively replicating parasites in the cell or by Brownian motion.

Minor concerns

1- Figure Ext Dat Fig 6a and Ext Dat Fig 8 do not show how efficiently the OMVs are labeled. The authors can ei Ext Dat Fig 6a ther use the NTA to do this (if there Nanosight has the filters for fluorescence detection). Labeled versus unlabeled vesicles can be quantified and differences can be measure. Otherwise, just remove the figures. They are not informative and does not show anything.

Response: Thank you for your suggestion. We have decided to remove the Extended Data Figure 6a and Extended Data Figure 8e from the manuscript.

2- Why is there so much DiL signal in the control? Please include the percentage of DiL positive parasites and statistical analysis of the measurements.

Response: DiI is a lipophilic membrane stain that diffuses laterally to stain the entire cell. It shows very weak background fluorescence unless incorporated into membranes. When the parasites uptake substantial DiI-stained OMVs, the DiI signal accumulates on the parasite membrane, leading to a strong fluorescent signal. After incubation with OMVs, all the parasites (as indicated by Hoechst staining) showed strong DiL signal and were subsequently killed (Extended Data Figure 7b). As suggested, we have added the measurements of DiI-positive parasites along with the fluorescence microscope images in Extended Data Figure 7b) as suggested (Figure below).

OMV incorporation and OMV induction assay of PC. Confocal images of *P. falciparum* 3D7 asexual parasites incubated for 20 min with DiI-labeled OMVs. The white arrowheads indicate healthy red blood cells (RBCs), the red arrowheads indicate parasitized cells (iRBC) that were lysed by Su_YN1 OMVs. Parasite nuclei were stained with Hoechst 33342 (Hst). BF, bright field. Scale bar, 5 μ m. The right panel presents statistical analysis of the DiI signal distribution in RBC (Hst negative, Hst-) and iRBC (Hst positive, Hst+) cells.

Reviewer #3 (Remarks to the Author):

This study shows further analysis of *Serratia ureilytica* Su_YN1, a bacterium previously identified by the authors as secreting the anti-*Plasmodium* lipase protein AmLip. AmLip is incorporated into the cell membrane of *Plasmodium* and kills *Plasmodium* by hydrolyzing lipids. They revealed how effectors such as AmLip are selectively transferred to *Plasmodium* via OMVs. The study is well-organized and provides new insights into the insect-bacteria-*Plasmodium* tripartite interaction.

The paper is worthy of publication in Nature Communications, but I would like to request a few points of clarification, as listed below.

Response: We are grateful to the Reviewer for these positive feedback.

(line 85)

Although the membrane vesicle-like structures in the midgut of blood-fed mosquitoes contain AmLip, there seems to be no direct evidence that these vesicles are of bacterial origin, in particular, that they are produced by Su_YN1. In vitro experiments using pre-cleared FBS confirmed that Su_YN1 can produce OMVs. It is recommended to prepare somehow vesicle-free blood and perform artificial blood-feeding experiments. (I understand that experiments using serum are difficult because mosquitoes hesitate to suck serum)

Response: Budding of OMVs from the bacteria can be seen in the gut of blood-fed mosquitoes (as depicted in Figure 1b, upper panel). To further investigate this phenomenon, we conducted cryo-electron microscopy on mosquito guts containing Su_YN1 bacteria and confirmed the occurrence of OMV biogenesis (Figure A below). These results demonstrate that OMV budding does occur in Su_YN1 in the mosquito gut. Additionally, as per the suggestion of the reviewer, we conducted an experiment feeding Su_YN1-carrying mosquitoes with EV-free FBS and observed OMVs in the mosquito gut using TEM. We observed OMVs only in the guts of Su_YN1- carrying mosquito fed with FBS, but not in axenic mosquito fed with FBS or Su_YN1-carrying mosquito fed with a sugar meal (Figure B below).

Detection of Su_YN1 OMVs in the mosquito gut. (A) Cryo-scanning electron microscopy image of the midguts of *An. stephensi* carrying Su_YN1 36 h after a blood meal. Blue arrow heads indicate Su_YN1 bacteria. Red arrow heads indicate outer membrane vesicles (OMVs). The scale bar represents 200 nm. (B) Transmission electron microscope (TEM) images of Su_YN1 OMVs from the mosquito midgut lumen. *Anopheles* mosquitoes carrying Su_YN1, or axenic *Anopheles* mosquitoes, were fed with either FBS or a sugar meal, and 36 hours later, their midguts were sectioned for TEM analysis. The white arrow heads indicate OMVs. Scale bar, 1 μ m.

We have also performed IEM on Su_YN1 OMVs from the mosquito gut section using the exosomal marker CD63 antibody (Cat# A22343, ABclonal) and AmLip antiserum. We did not detect any positive signals of CD63 on Su_YN1 OMVs, whereas AmLip was detected (Figure below). The lack of the exosomal marker CD63 positive staining and the presence of AmLip positive staining indicate that these vesicle-like structures are not of eukaryotic origin. We have included this data in Supplementary Fig. 1c in the revised manuscript.

IEM of Su_YN1 OMVs in the mosquito gut. We used immune electron microscopy (IEM) to detect CD63 and the lipase AmLip on the OMVs of Su_YN1 in the midgut of *Anopheles* mosquitoes. An ultrathin cryosection of the midgut was analyzed, and positive staining was indicated by white arrowheads. The scale bar is 100 nm.

This is a comment related to the previous report by the authors published in Nature Microbiology. Is there any difference in the ability of AmLip alone (recombinant protein) and OMVs in which AmLip is incorporated to inhibit *Plasmodium* ookinete formation?

Response: Both AmLip alone (recombinant protein) and OMVs-bind AmLip inhibit *Plasmodium* parasites. However, the interaction of lipase AmLip with OMVs is required for robust anti-*Plasmodium* activity. In the absence of OMVs, AmLip protein diffuses upon release and slowly targets the *Plasmodium* cells. Consequently, inhibiting *Plasmodium* parasites with AmLip alone requires an IC50 concentration exceeding 2 µg/ml and a 6-hour incubation. However, binding to OMVs significantly enhances the malaria targeting of AmLip and promotes its anti-*Plasmodium* activity. AmLip-containing OMVs require an IC50 concentration of around 5 µg/ml (the concentration of the total protein content of OMVs) and just a few minutes of incubation. Notably, AmLip only accounts for less than 5% of the total proteins in OMVs. Thus, binding to OMVs strongly enhances the malaria targeting and anti-*Plasmodium* activity of AmLip.

In each of the experiments in Figure 3, what would happen if AmLip protein alone was used without OMVs? Based on their findings, the presence of OMVs should promote the efficiency of AmLip in killing *Plasmodium*.

Response: As per our response to the above comment, the interaction of lipase AmLip with OMVs is required for robust anti-*Plasmodium* activity. AmLip alone does not kill *Plasmodium* cells so efficiently. Binding to OMVs significantly enhances the malaria targeting of AmLip and promotes its anti-*Plasmodium* activity. Thus, binding to OMVs strongly promote the efficiency of AmLip in killing *Plasmodium*.

What is the possible mechanism by which OMVs can invade and destroy infected erythrocytes while not affecting normal erythrocytes? Please provide the readers with possible hypotheses.

Response: The targeting of OMVs to *Plasmodium* infected RBC (iRBC) is a parasite-specific mechanism. *Plasmodium*-infected RBC strongly uptake lipid from the medium while healthy RBC does not. At asexual stages, the parasites are concealed within the host RBC, and the membrane of iRBC is dramatically modified by parasites, which export proteins to modify the host RBC membrane (*Semin Hematol.* 2004 Apr;41(2):173-88.; *PNAS*, April 27, 2015, 112 (19) 6068-6073). The PC scavenge pathway proteins, in particular, are incorporated in the iRBC membrane in this way (*J Cell Biol.* 1989 Jun 1; 108(6): 2183–2192). Thus, the iRBC membrane is distinct from the healthy RBC membrane and could recruit OMVs for internalization. We added additional comments on the selective entering mechanism in the discussion: “The active PC scavenging of parasitized RBCs distinct from healthy RBCs, which do not uptake OMVs and are exempt from being lysed by the OMV cargo AmLip.”. Please refer to the revised manuscript.

What do you think is the advantage of midgut bacterium species such as Su_YN1 having the mechanism of OMVs production found by the authors?

Response: Our ongoing study indicates that the ability to produce OMVs provides an advantage for Su_YN1 to adapt to the harsh gut microenvironment of blood-fed mosquitoes.

Peer Review comments, second round -

Reviewer #1 (Remarks to the Author):

I think that the authors have performed additional experiments, which have strengthened the manuscript. I have no further comments.

Reviewer #2 (Remarks to the Author):

The reviewer appreciates the efforts of the authors to revise the manuscript, however, some of the experiments presented as response were not performed correctly and therefore cannot be accepted as appropriately addressing the comments from the reviewers.

Major concern

1- As reviewer 1 correctly pointed out, EVs are secreted by eukaryotic cells, this includes mosquito cells. The supplementary figure 1c presents the lack of immunolabeling of the vesicles with CD63. However, the authors used a monoclonal antibody. This means that the antibody likely only recognizes an epitope within the protein in the human version of the protein. This antibody has been validated to be used in human cells/tissues, rats, and mice. This does not mean that the antibody recognizes mosquito CD63. The authors should show that the antibody does in fact cross react with mosquito tissues and mosquito EVs. That there are not EVs labeled by CD63 is concerning, as mosquito cells are known to produce vesicles and some EVs would be expected to be mosquito derived.

2- No details about the proteomic analysis are provided. The raw data is not available and tables from the proteomic analysis. The data presented are only the most abundant proteins, however, the authors are cherry picking by only showing the highest abundant proteins. All the proteins detected should be provided specially since an excel export file is easy to create from proteomics data. The raw data should be deposited in a publicly available database.

3- Figure 4C is confusing, since OMV+ is only added above the infected RBCs, the reviewer assumed that the uninfected RBC was not treated. Furthermore, these data should be validated by time lapse microscopy as done with the infected RBC.

4- The lack of FBS does affect the fitness of the bacterium as shown in growth curve requested by reviewer 1. How can the authors be sure that the lower numbers of OMVs are not due to lower numbers of bacteria and stressed-out bacteria? Was the NTA analysis presented in figure 1d done from the same volume of bacteria or the same number of bacteria? In the materials and methods, they describe using 26 ml, which means that in fact less bacteria were used for the FBS-. Thus, this adds error to the analysis. Given this, the experiments should be performed using the same number of bacteria. Was the BCA done in the same volume of cultured bacteria as well? This would explain the discrepancy between the RNA and the protein levels as was my concern with the model. The qRT-PCR was done with 2 OD of bacteria. Therefore, no induction was observed. I am concerned that you made an error throughout the experiments and therefore should be repeated. Small RNA regulation of the protein expression may be due to the stress that the bacteria are under in the FBS-. Regulation of the mRNA and the protein should also be validated in the midgut of the mosquito as factors within the mosquito are likely to maintain the bacteria. The authors explain that they values were normalized to the OD600. How was this normalization done? Why not use the number of bacteria as done with the qRT-PCR?

5- DiL is red, the rebuttal letter has the colors switched. If the labeling with DiL was performed properly, the vesicles should be all red. Labeling with DiL can be done with OMVs only without the need to label the bacteria. This can, and should, be validated with NTA to quantify the number of vesicles that are not properly labeled. What you are presenting in Dot #1 may be proteins. The stain is incorporated in the lipid membrane and occurs without the need to label the cell of origin. How were the OMVs labeled with the AmLip? Description of the immunolabeling in the EM figures

described the use of Tween 20 to label with the antiserum. This would have removed the lipids and affected integrity of the vesicles. Please add details to describe the labeling of AmLip within the OMVs.

6- The figure "OMV incorporation and OMV induction assay of PC" panel B should be incorporated into the manuscript and not only in the rebuttal. These results indicate that the source is likely to be lipoproteins and the protein portion may be important for uptake of the PC. You also seem to have two populations of NBD-PC labeled particles.

Reviewer #3 (Remarks to the Author):

The authors have well addressed the revisions suggested by the reviewers and the manuscript can be recommend for publication.

Manuscript: NCOMMS-22-45996-A

Title: Outer membrane vesicles from a mosquito commensal mediate targeted killing of *Plasmodium* parasites via the phosphatidylcholine scavenging pathway

We appreciate all reviewers for the evaluation of our manuscript, and for the additional comments and suggestions by reviewer 2#. We have carefully revised the manuscript taking into account of the reviewers' comments. Our point-by-point responses are provided below. Please note that reviewers' comments are quoted in bold and our responses follow in plain text.

REVIEWER COMMENTS

Reviewer #1 (Remarks to the Author):

I think that the authors have performed additional experiments, which have strengthened the manuscript. I have no further comments.

Response: This comment is much appreciated.

Reviewer #2 (Remarks to the Author):

The reviewer appreciates the efforts of the authors to revise the manuscript, however, some of the experiments presented as response were not performed correctly and therefore cannot be accepted as appropriately addressing the comments from the reviewers.

Major concern

1- As reviewer 1 correctly pointed out, EVs are secreted by eukaryotic cells, this includes mosquito cells. The supplementary figure 1c presents the lack of immunolabeling of the vesicles with CD63. However, the authors used a monoclonal antibody. This means that the antibody likely only recognizes an epitope within the protein in the human version of the protein. This antibody has been validated to be used in human cells/tissues, rats, and mice. This does not mean that the antibody recognizes mosquito CD63. The authors should show that the antibody does in fact cross react with mosquito tissues and mosquito EVs. That there are not EVs labeled by CD63 is concerning, as mosquito cells are known to produce vesicles and some EVs would be expected to be mosquito derived.

Response: We purchased additional CD9 antibodies specifically designed to recognize eukaryotic EVs. We have two antibodies: the anti-CD63 rabbit monoclonal antibody (Cat# A22343, ABclonal) and the anti-CD9 mouse monoclonal antibody (Santa Cruz Cat#sc-59140).

The latter is widely used as exosome biomarker (Kumar, A., et al., *The polysaccharide chitosan facilitates the isolation of small extracellular vesicles from multiple biofluids. J Extracell Vesicles*, 2021. 10(11): p. e12138. Lättekivi, F., et al., *Profiling Blood Serum Extracellular Vesicles in Plaque Psoriasis and Psoriatic Arthritis Patients Reveals Potential Disease Biomarkers. Int J Mol Sci*, 2022. 23(7):4005. Liu, T., et al., *Evaluating adipose-derived stem cell exosomes as miRNA drug delivery systems for the treatment of bladder cancer. Cancer Med*, 2022. 11(19): 3687-3699.) and has been reported to also recognize mosquito exosomes (Reyes-Ruiz, J.M., et al., *Isolation and characterization of exosomes released from mosquito cells infected with dengue virus. Virus Research*, 2019. 266: 1-14.). We used both antibodies and verified their ability to recognize human and mosquito CD63 or CD9 (refer to Figure A below). The slight variation in molecular weight observed may be attributed to species-specific differences in the proteins. Notably, when utilizing the new CD9 antibody, we also observed no positive staining on the vesicles in mosquito gut sections (Figure B below). This finding further confirms that the vesicles are not secreted by eukaryotic cells. Supplementary Figure 1c and d in the revised manuscript have been updated to incorporate these results.

Detection of CD63 or CD9 in mosquito, human and *Serratia* bacteria samples. (A) Western blot detection of CD63 and CD9 using 30 μ g of total protein extracts obtained from *Anopheles* MSQ43 cells and human 293T cells. The left panel shows Coomassie blue staining of the samples as a loading control. The arrowheads indicate the target bands. (B) Detection of CD9 on OMVs by immune electron microscopy (IEM) in an ultrathin cryosection of the midgut of Su_YN1-carrying *Anopheles* mosquito. Scale bar, 100 nm.

2- No details about the proteomic analysis are provided. The raw data is not available and tables from the proteomic analysis. The data presented are only the most abundant proteins, however, the authors are cherry picking by only showing the highest abundant proteins. All the proteins detected should be provided specially since an excel export file is easy to create from proteomics data. The raw data should be deposited in a publicly available database.

Response: We have included the comprehensive details of OMV proteomic analysis in the revised manuscript's Methods section. The original Supplementary Fig.2 has been removed, and the complete list of Su_YN1 OMV proteomic analysis is now provided as Supplementary Table 1. The mass spectrometry proteomics data associated with this study have been deposited to the ProteomeXchange Consortium via the PRIDE partner repository. The dataset identifier is PXD042831, and the Project Name is "*Serratia* bacteria Su_YN1 OMV LC-MSMS". Once the paper is published, the data will be publicly available. Reviewers can access the data using the following reviewer account credentials: Username: reviewer_pxd042831@ebi.ac.uk; Password: DYP5gE6t.

3- Figure 4C is confusing, since OMV+ is only added above the infected RBCs, the reviewer assumed that the uninfected RBC was not treated. Furthermore, these data should be validated by time lapse microscopy as done with the infected RBC.

Response: Please note that the uninfected RBCs in Figure 4C were also treated with OMVs. The RBCs were treated with OMVs, washed with PBS to remove any unbound OMVs, and then fixed for SEM analysis. Since uninfected RBCs do not bind OMVs, no OMVs are observed in uninfected RBCs. To provide additional evidence, we present time-lapse microscopy images of uninfected RBCs treated with DiI-stained OMVs (see Figure below). These images are included as Extended Data Figure 6b, corresponding to Figure 4b.

Time-lapse microscopy images of uninfected RBC treated with DiI-stained OMVs. Confocal live tracking was performed to observe the uptake of DiI-labeled OMVs by uninfected red blood cell (RBC). Images were captured at 30 sec intervals for a duration of 10 min following the addition of DiI-labeled OMVs to the RBC. BF, bright field. Scale bar, 5 μ m.

4- The lack of FBS does affect the fitness of the bacterium as shown in growth curve requested by reviewer 1. How can the authors be sure that the lower numbers of OMVs are not due to lower numbers of bacteria and stressed-out bacteria? Was the NTA analysis

presented in figure 1d done from the same volume of bacteria or the same number of bacteria? In the materials and methods, they describe using 26 ml, which means that in fact less bacteria were used for the FBS-. Thus, this adds error to the analysis. Given this, the experiments should be performed using the same number of bacteria. Was the BCA done in the same volume of cultured bacteria as well?

Response: Su_YN1 OMVs are specifically induced by serum, and the lower numbers of OMVs are not due to lower numbers of bacteria under FBS- conditions. To address concerns regarding OMV quantification, we have taken into account the same number of bacteria in both OMV quantification assays and NTA assay for accurate assessment. This approach allows us to normalize OMV production to 10^9 bacteria, as commonly practiced in quantifying OMV biogenesis under varying growth conditions or between different bacterial strains (*Shan, Q., M. Dwyer, and M. Gadjeva, Distinct Susceptibilities of Corneal Pseudomonas aeruginosa Clinical Isolates to Neutrophil Extracellular Trap-Mediated Immunity. Infection and immunity, 2014. 82. Roier, S., et al., A novel mechanism for the biogenesis of outer membrane vesicles in Gram-negative bacteria. Nature Communications, 2016. 7(1): 10515.*). To further address the reviewer's concerns, we conducted additional experiments by culturing Su_YN1 under FBS- and FBS+ conditions, and collected the culture supernatant for OMV quantification once the bacterial density reached 1OD and 2OD in each group, respectively. The results demonstrated that Su_YN1 produces significantly more OMVs under FBS induction compared to the same cell densities under FBS- conditions (refer to the Figure below). These results have been added as Extended Data Fig.2d.

OMVs quantification assay of Su_YN1 at various cell densities. OMVs of Su_YN1 cultured under FBS- and FBS+ conditions were quantified once the bacterial density reached 1OD and 2OD in each group, respectively. The results are presented as mean \pm SD (n=3). Statistical significance was determined using a two-tailed Student's t-test.

This would explain the discrepancy between the RNA and the protein levels as was my

concern with the model. The qRT-PCR was done with 2 OD of bacteria. Therefore, no induction was observed. I am concerned that you made an error throughout the experiments and therefore should be repeated. Small RNA regulation of the protein expression may be due to the stress that the bacteria are under in the FBS-. Regulation of the mRNA and the protein should also be validated in the midgut of the mosquito as factors within the mosquito are likely to maintain the bacteria. The authors explain that they values were normalized to the OD600. How was this normalization done? Why not use the number of bacteria as done with the qRT-PCR?

Response: We have found that the transcription of *AmLip* is not significantly affected by serum, indicating that serum does not have a direct impact on *AmLip* gene expression at the transcriptional level. However, we have discovered that the expression of *AmLip* is post-transcriptionally regulated by a sRNA. This sRNA responses to serum induction and regulates the translation of *AmLip* mRNA. The intricate regulation of *AmLip* mRNA and protein by this sRNA will be published in another paper. In our qRT-PCR experiments, we ensured that an equal number of bacteria was collected for RNA extraction. Specifically, in each group, we collected the same amount of bacteria (total 2 OD, equivalent to 2×10^9 bacterial cells) at various time points. We have made revisions to the Materials and methods section to clarify these details and avoid any potential misunderstandings.

5- DiI is red, the rebuttal letter has the colors switched. If the labeling with DiI was performed properly, the vesicles should be all red. Labeling with DiI can be done with OMVs only without the need to label the bacteria. This can, and should, be validated with NTA to quantify the number of vesicles that are not properly labeled. What you are presenting in Dot #1 may be proteins. The stain is incorporated in the lipid membrane and occurs without the need to label the cell of origin. How were the OMVs labeled with the *AmLip*? Description of the immunolabeling in the EM figures described the use of Tween 20 to label with the antiserum. This would have removed the lipids and affected integrity of the vesicles. Please add details to describe the labeling of *AmLip* within the OMVs.

Response: We apologize for the confusion caused by the incorrect labeling in the rebuttal letter, and we appreciate the reviewer for bringing this to our attention. We have now corrected the two labels for *Amlip* and OMV in the figure. We would like to clarify that the vesicles stained with DiI (not DiL) appear as red fluorescence (as shown in the Figure below).

Regarding Extended data fig.3f, we would like to clarify that the image represents purified OMVs attached to a cover slip and does not contain bacterial cells. The OMVs were pre-stained

with DiI, purified, and washed to remove any free DiI dye. The stained OMVs were then resuspended in PBS and incubated with a Poly-D-lysine treated cover slip to facilitate the attachment of OMVs to the cover slip surface before immunofluorescence staining. The OMVs attached to the cover slip were then immunolabeled with AmLip antiserum and Alexa 488-conjugated goat anti-mouse IgG. This procedure follows established protocols used by other researchers in vesicle-protein detection experiments (*Bananis, E., et al., Microtubule and motor-dependent endocytic vesicle sorting in vitro. J Cell Biol, 2000. 151(1): 179-86.* *Jiang, M., et al., Reductions in bacterial viability stimulate the production of Extra-intestinal Pathogenic Escherichia coli (ExPEC) cytoplasm-carrying Extracellular Vesicles (EVs). PLoS Pathog, 2022. 18(10): e1010908.*). Detailed information regarding this procedure has been included in the revised manuscript's Method section to provide a clear description of the AmLip labeling with the OMVs.

In the figure, Dot #1 represents the OMV as it carries the DiI signal. The weak signal observed may be due to insufficient staining. However, we regret to inform you that we are currently unable to conduct a dual-fluoresce assay of the vesicles as our NanoSight NS300 nano-tracking system (Malvern Panalytical) and the nanoflow cytometry measurement facility (NFCM) (by Neoland BioSciences Co.,Ltd) do not have both fluoresce channels required for such analysis. Regarding the immunofluorescence assay of OMVs in Extended data fig.3f, we would like to clarify that it was not treated with Tween 20. However, the Immune electron microscopy (IEM) experiments did involve Tween 20 treatment. We believe that the use of Tween 20 (which is milder compared to other polysorbates such as Tween 80) in the IEM protocol does not significantly affect the integrity of the vesicles or remove membrane lipids. Similar treatments with Tween 20 are commonly adopted by other researchers in IEM experiments for detecting of membrane-based structures (*Mattosco, D., et al., Immunogold Electron Microscopy of the Autophagosome Marker LC3. Bio Protoc, 2017. 7(24): p. e2648.* *Yuan, Y., M. Li, and Y. Hong, Light and electron microscopic analyses of Vasa expression in adult germ cells of the fish medaka. Gene, 2014. 545(1): p. 15-22*). Our colleagues have also successfully used Tween-20 treatment for detecting plant vesicles. We appreciate the reviewer's attention to these details, and we apologize for any confusion caused.

Colocalization of AmLip with OMVs. Confocal images showing the colocalization of AmLip (green channel) and OMVs (DiI stained, red channel). The signals are quantified in the right panel.

6- The figure “OMV incorporation and OMV induction assay of PC” panel B should be incorporated into the manuscript and not only in the rebuttal. These results indicate that the source is likely to be lipoproteins and the protein portion may be important for uptake of the PC. You also seem to have two populations of NBD-PC labeled particles.

Response: We thank reviewer for the suggestion. We have included the figure in the revised manuscript as Supplementary Figure 4d. The quantification has been normalized to 10^9 bacteria. Regarding the dispersive distribution of NBD-PC labeled particles, we speculate that it may reflect the varying extent of PC incorporation into OMVs. However, at present, we do not have a complete understanding of how serum PC is incorporated by the bacteria to generate OMVs. We are actively investigating the underlying mechanisms in our ongoing work. We have added a discussion on this topic in the revised manuscript.

Reviewer #3 (Remarks to the Author):

The authors have well addressed the revisions suggested by the reviewers and the manuscript can be recommend for publication.

Response: This comment is much appreciated.

Peer Review comments, third round -

Reviewer #2 (Remarks to the Author):

The authors have performed all the requested changes and no further changes are required.

Manuscript: NCOMMS-22-45996-T

Title: Outer membrane vesicles from a mosquito commensal mediate targeted killing of *Plasmodium* parasites via the phosphatidylcholine scavenging pathway

We appreciate all reviewers for the evaluation of our manuscript. Our point-by-point response is provided below. Please note that reviewers' comments are quoted in bold and our responses follow in plain text.

REVIEWER COMMENTS

Reviewer #2 (Remarks to the Author):

The authors have performed all the requested changes and no further changes are required.

Response: This comment is much appreciated.